# Trajectory Seriation via Spectral Tangent Alignment and Global Embedding

## Abstract

This paper addresses the problem of linear seriation: recovering the intrinsic order of noisy samples drawn from an unknown one-dimensional manifold embedded in a higher-dimensional space. We propose a multi-stage approach that first robustly estimates local tangent directions using Principal Component Analysis (PCA) on neighborhoods, establishing theoretical consistency for these local estimates. Global orientation consistency of these tangents is then achieved through a spectral relaxation of a pairwise alignment objective. Finally, a globally consistent 1D embedding is computed by solving a carefully formulated linear system (or equivalently, a spectral problem on a derived Laplacian) that aligns the embedding with the oriented local projections. This method effectively leverages local geometric information while ensuring global coherence, producing an ordering robust to noise, curvature, and initial data rotation. We demonstrate its performance on simulated manifold data and discuss the theoretical underpinnings of its core components.

## 1 Problem Formulation

We address the problem of linear seriation: recovering the intrinsic order of points sampled from a one-dimensional manifold (a curve) embedded in a higher-dimensional Euclidean space. Let $\mathcal{M} \subset \mathbb{R}^d$ be an unknown smooth one-dimensional manifold. We assume $\mathcal{M}$ can be parameterized by an injective function $\gamma : [0, 1] \to \mathbb{R}^d$, where the parameter $\theta \in [0, 1]$ represents the intrinsic order along the curve. We further assume the curve has non-zero speed, i.e., $\|\gamma'(\theta)\|_2 > 0$ for all $\theta \in [0, 1]$. The points $\gamma(0)$ and $\gamma(1)$ define the manifold's endpoints.

We are given a set of $n$ observations, denoted $\mathscr{X}_n = \{x_1, x_2, \ldots, x_n\} \subset \mathbb{R}^d$. Each observation $x_i$ is generated from an unknown underlying parameter $\theta_i \in [0, 1]$ and corrupted by additive noise:

$$x_i = \gamma(\theta_i) + \epsilon_i, \quad i = 1, \ldots, n. \tag{1}$$

Here, the parameters $\{\theta_i\}_{i=1}^n$ are distinct and their values are unknown. The noise terms $\{\epsilon_i\}_{i=1}^n$ are assumed to be i.i.d. zero-mean sub-Gaussian random vectors in $\mathbb{R}^d$.

The objective of linear seriation is to estimate the unknown ordering of these latent parameters $\{\theta_i\}$. Since the inherent directionality of the manifold (i.e., which endpoint corresponds to $\theta = 0$ versus $\theta = 1$) is typically not identifiable from the data $\mathscr{X}_n$ alone, we aim to find a permutation $\phi$ of the indices $\{1, \ldots, n\}$ such that the parameters indexed by $\phi$ are consistently ordered, i.e., either

$$\theta_{\phi(1)} < \theta_{\phi(2)} < \cdots < \theta_{\phi(n)} \quad \text{or} \quad \theta_{\phi(1)} > \theta_{\phi(2)} > \cdots > \theta_{\phi(n)}. \tag{2}$$

Alternatively, the goal can be viewed as assigning a real-valued score $s(x_i)$ to each observation $x_i$ such that the order of these scores reflects the true order of the underlying parameters $\{\theta_i\}$, or its reverse. That is, $s(x_i)$ should be monotonically increasing or monotonically decreasing with respect to the true parameter values $\theta_i$.

## 2 Overview of Related Work and Our Method

The problem of seriation, which seeks to recover a latent ordering among a set of items based on pairwise measurements, has a rich history and diverse applications. Originating in archaeology for the chronological ordering of artifacts Robinson (1951), its principles are now applied in fields such

as sparse matrix computations for envelope reduction Barnard et al. (1993), bioinformatics for tasks like *de novo* genome assembly Garriga et al. (2008); Meidanis et al. (1998); Recanati et al. (2017), and network analysis for time synchronization Elson et al. (2004).

Many existing seriation methods operate on a pre-defined or computed $n \times n$ matrix $A$, representing pairwise similarities (or dissimilarities) between $n$ items. The underlying assumption is often that this matrix, when items are correctly ordered, exhibits a specific structure, most notably the Robinson property. A matrix has Robinson property if whenever we move away from the diagonal along any row or column, the values decrease (cf. Definition 1.1 and 1.2 in Recanati et al. (2018) and Fogel et al. (2013)). The challenge then lies in finding the permutation $\phi$ that transforms the observed, permuted similarity matrix into (or close to) this ideal Robinson form. **However, when the original data consists of points in a metric space (e.g., $\mathbb{R}^d$), deriving such a global similarity matrix can lead to a loss of finer geometric information inherent in the point cloud coordinates (cf. Appendix A).**

### 2.1 Spectral and Affinity-Based Methods for Seriation

A widely adopted class of methods for seriation from such similarity matrices are spectral techniques. These methods typically construct a graph Laplacian from the similarity matrix $A$. For linear seriation, the seminal work by Atkins et al. Atkins et al. (1998) demonstrated that the Fiedler vector—the eigenvector corresponding to the second smallest eigenvalue of the graph Laplacian $L_A = \text{diag}(A\mathbf{1}) - A$—provides an ordering of the items. Sorting the components of this Fiedler vector yields the estimated seriation, known to exactly recover the order for ideal Robinson matrices. Extensions of spectral methods also address circular seriation. For instance, Coifman et al. Coifman et al. (2008) utilized the first two non-trivial eigenvectors of a normalized graph Laplacian to embed data points onto a circle, with the ordering then determined by their angles. The theoretical underpinnings of these methods often connect the discrete graph Laplacian to continuous Laplace-Beltrami operators on underlying manifolds Belkin & Niyogi (2003); Coifman & Lafon (2006); Singer (2006). Other dimensionality reduction techniques like Multidimensional Scaling (MDS) Kruskal & Wish (1978) and kernel PCA Schölkopf et al. (1997) have also been explored for embedding data in ways that might preserve latent ordering. An overview of spectral clustering, which shares methodology, can be found in Von Luxburg (2007).

### 2.2 Statistical Approaches and Robustness in Noisy Seriation

Real-world data often presents noisy similarity measurements, where the observed matrix $A$ is a perturbation of an ideal structured matrix $F_\phi$. While spectral methods have proven effective, their robustness to noise, especially when $F_\phi$ deviates from simple structures, has been a subject of investigation Giraud et al. (2023); Natik & Smith (2021). Recent research has focused on developing algorithms with provable guarantees in noisy settings, analyzing specific matrix classes Cai & Ma (2023), or studying seriation within latent space models Janssen & Smith (2022). These analyses often aim to characterize error rates Flammarion et al. (2019) and establish conditions for reliable recovery of the latent permutation $\phi$.

### 2.3 Our Approach: Geometric Seriation Directly from Point Cloud Data

**Motivated by the potential information loss in methods relying solely on pre-computed global similarity matrices, our work, STAGE (Spectral Tangent Alignment and Geometric Embedding), directly leverages the geometric structure of the original observed data points $\{x_i\}_{i=1}^n \subset \mathbb{R}^d$.** Instead of first collapsing the point cloud information into a global similarity matrix, our method proceeds by:

1. **Estimating Local Manifold Geometry:** We employ local Principal Component Analysis (PCA) within neighborhoods of each data point $x_i$ to estimate the local tangent direction $\widehat{v}_i$ of the underlying 1D manifold. This approach is inspired by techniques in manifold learning that infer local structure from point coordinates Roweis & Saul (2000).

2. **Global Orientation of Tangents:** A spectral method is then used to consistently orient these local tangent vectors across the manifold, yielding $\widehat{s}_i \widehat{v}_i$.

3. **Constructing a Geometrically-Informed Global Embedding:** The core of our distinction lies in using these oriented local geometric features ($\widehat{s}_k \widehat{v}_k$) and the original point coordinates

$(x_k)$ to directly formulate the objective for a global 1D embedding $y$. This is achieved by solving a least-squares problem, such as minimizing $\sum_{i,j:\|x_i-x_j\|_2 \leq r}(y_j - y_i - \widehat{W}_{ij})^2$, where $\widehat{W}_{ij} = \frac{1}{2}(\widehat{s}_i\widehat{v}_i + \widehat{s}_j\widehat{v}_j)^{\mathsf{T}}(x_j - x_i)$.

This direct integration of local, coordinate-based geometric estimates into the global ordering objective allows our method to be more adaptive to the specific geometry of the data manifold. While many manifold learning techniques (e.g., Laplacian Eigenmaps Belkin & Niyogi (2003), Diffusion Maps Coifman & Lafon (2006)) also start from point coordinates, they typically first construct a global similarity graph (e.g., using a heat kernel $W_{kl} = \exp(-\|x_k - x_l\|^2/t)$) and then apply spectral methods to this graph. Our approach defers the aggregation of local information until the formulation of the final embedding objective, potentially preserving finer geometric details relevant for accurate seriation. The work by Recanati et al. Recanati et al. (2018) also explores enhancing latent orderings using multidimensional Laplacian embeddings and refining similarities based on local fits within the resulting embedding. Issartel et al. Issartel et al. (2024) provide a minimax optimal algorithm operating on noisy similarity matrices by aggregating local bisections. The robustness of our method stems from the local averaging inherent in PCA and the global consistency enforced by the spectral orientation and the final embedding stage.

## 3 THE STAGE ALGORITHM

Let $G_r(\mathscr{X}_n)$ denote the standard $r$-neighborhood (a.k.a. $\varepsilon$-) graph of the sample $\mathscr{X}_n$, where

$$i \sim j \iff \|x_i - x_j\| \leq r.$$

We let $A = (A_{ij}) = 1\{i \sim j\}$ be the adjacency matrix of $G_r(\mathscr{X}_n)$ and $L = D-A$, its (combinatorial) Laplacian. We let $\lambda_2(L)$ denote the smallest nonzero eigenvalue of $L$, a.k.a. algebraic connectivity of the graph, and let $d_i = \sum_j A_{ij}$ denote the degree of node $i$.

Our method proceeds in four stages: first, estimate local tangent vector at each point, using its neighbors in $G_r(\mathscr{X}_n)$; second orient them globally using a spectral method; third, compute a 1D embedding that aligns with these oriented local structures via least squares; fourth, derive the final order from the embedding. Below, we detail each step.

### 3.1 LOCAL TANGENT ESTIMATION

The first stage of our algorithm estimates the local tangent direction $\widehat{v}_i \in \mathbb{R}^d$ at each point $x_i \in \mathscr{X}_n$. This is achieved by performing Principal Component Analysis (PCA) on the set of its neighbors within the $r$-neighborhood graph $G_r(\mathscr{X}_n)$.

For each point $x_i \in \mathscr{X}_n$, let $d_i$ be its degree in $G_r(\mathscr{X}_n)$ (i.e., the number of points $x_j$ such that $\|x_j - x_i\|_2 \leq r$). If $d_i < d_{\min}$ (where $d_{\min} \geq 2$ is a minimum required number of neighbors, e.g., $d_{\min} = 2$ or slightly higher for stability), the local tangent cannot be reliably estimated for $x_i$. For all other points, we compute:

1. The local sample mean:

$$\widehat{\mu}_i = \frac{1}{d_i} \sum_{j:j\sim i} x_j. \tag{3}$$

2. The local sample covariance matrix:

$$\widehat{\Sigma}_i = \frac{1}{d_i} \sum_{j:\, j\sim i} (x_j - \widehat{\mu}_i)(x_j - \widehat{\mu}_i)^{\mathsf{T}}. \tag{4}$$

3. The estimated local tangent direction, $\widehat{v}_i \in \mathbb{R}^d$, is the (normalized) eigenvector corresponding to the largest eigenvalue of $\widehat{\Sigma}_i$.

The vector $\widehat{v}_i$ represents the direction of maximum variance in the local neighborhood of $x_i$ and thus serves as an approximation of the tangent vector to the underlying manifold $\mathcal{M}$ at or near $x_i$.

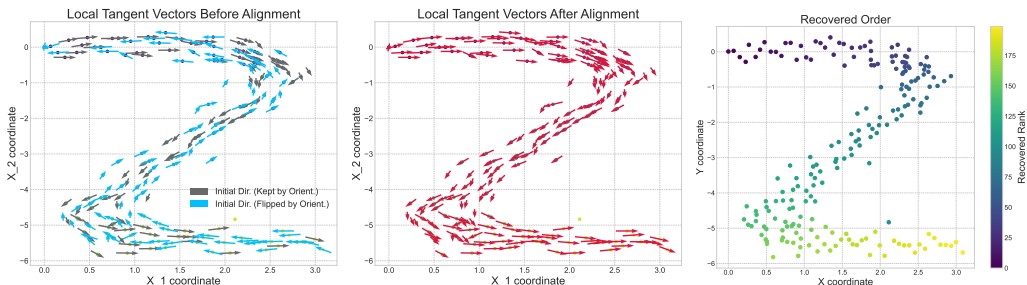

Figure 1: (Left) Initial PCA-derived tangent vectors. Their orientations are arbitrary before alignment. Colors indicate their fate after the orientation step: gray for vectors whose initial direction is kept, and blue for those that will be flipped. (Middle) Tangent vectors after spectral orientation. The vectors are now globally aligned along the manifold's intrinsic direction. (Right) Recovered order shown by color.

**Remark 1** (Choice of neighborhood graph). *In practice, $r$-neighborhood graphs and $k$-nearest-neighbor graphs perform similarly. However, the $r$-neighborhood graph has a symmetric adjacency matrix, which makes it easier to analyze. The hyperparameters $k$ and $r$ influence performance, much like the perplexity parameter in t-SNE—there is no universally optimal choice. We require $k$ or $r$ to be large enough to ensure the graph is connected. Figures 7 and 8 in the appendix indicate that neighborhood size (measured by $k$) is not crucial for our algorithm's performance. Under identical settings, our method consistently outperforms t-SNE.*

## 3.2 Orientation Alignment

The local tangent vectors $\widehat{v}_i$ estimated in the previous step are unique only up to their sign, as $-\widehat{v}_i$ is also a valid principal eigenvector if $\widehat{v}_i$ is. To construct a meaningful global embedding, these local directions must be consistently oriented relative to each other across the manifold. Figure 1 (left) illustrates a set of such unaligned raw tangent vectors.

To achieve this global sign consistency, we seek a vector of signs $s = (s_1, \ldots, s_n)^\intercal$, where each $s_i \in \{-1, 1\}$, that maximizes the sum of dot products between signs of adjacent oriented tangents:

$$\max_{s \in \{-1,1\}^n} \sum_{i,j:\ i \sim j} s_i s_j \langle \widehat{v}_i, \widehat{v}_j \rangle. \tag{5}$$

This objective aims to make $s_i \widehat{v}_i$ and $s_j \widehat{v}_j$ point in similar directions if $\langle \widehat{v}_i, \widehat{v}_j \rangle > 0$, and opposite directions if $\langle \widehat{v}_i, \widehat{v}_j \rangle < 0$. The problem in (5) is equivalent to maximizing $s^\intercal \widehat{Q} s$, where $\widehat{Q}$ is a symmetric affinity matrix with entries

$$\widehat{Q}_{ij} = \begin{cases} \langle \widehat{v}_i, \widehat{v}_j \rangle & \text{if } i \sim j \\ 0 & \text{otherwise.} \end{cases} \tag{6}$$

Since optimizing over $s \in \{-1, 1\}^n$ is an NP-hard problem (related to Max-Cut), we employ a spectral relaxation. For any feasible $s$ in (5), we have $\|s\|_2^2 = n$. Relaxing the constraint $s_i \in \{-1, 1\}$ to $\|s\|_2^2 = n$ (i.e., $s$ lies on the sphere of radius $\sqrt{n}$), the problem becomes:

$$\widehat{u} = \arg \max_{s \in \mathbb{R}^n : \|s\|_2^2 = n} s^\intercal \widehat{Q} s. \tag{7}$$

The solution $\widehat{u}$ to this relaxed problem is the principal eigenvector of $\widehat{Q}$ (the eigenvector corresponding to the largest eigenvalue). The optimal signs for the original problem are then approximated by:

$$\widehat{s}_i = \text{sign}(\widehat{u}_i), \quad \text{for } i = 1, \ldots, n. \tag{8}$$

This procedure yields the consistently oriented tangent vectors $\widehat{s}_i \widehat{v}_i$. Figure 1 (right) shows these tangent vectors after the alignment process.

### 3.3 GLOBAL EMBEDDING VIA LEAST SQUARES

Having obtained consistently oriented local tangent vectors $\widehat{s}_i\widehat{v}_i$, we now seek a global 1-dimensional embedding $y = (y_1, \ldots, y_n)^\intercal \in \mathbb{R}^n$. The goal is that for any pair of neighboring points $(x_i, x_j)$ (i.e., $i \sim j$), the difference $y_j - y_i$ in the embedding space should approximate the estimated "signed distance" or progression along the manifold from $x_i$ to $x_j$.

A local estimate for this signed progression can be derived by projecting the displacement vector $x_j - x_i$ onto the local tangent at $x_i$, giving $\widehat{s}_i\widehat{v}_i^\intercal(x_j - x_i)$. Similarly, from $x_j$'s perspective, it would be $\widehat{s}_j\widehat{v}_j^\intercal(x_j - x_i)$. To create a symmetric and robust local target for $y_j - y_i$, we average the two oriented tangent vectors and project the displacement onto this average direction. This defines our target local increment $\widehat{W}_{ij}$:

$$\widehat{W}_{ij} := \frac{1}{2}(\widehat{s}_i\widehat{v}_i + \widehat{s}_j\widehat{v}_j)^\intercal(x_j - x_i), \quad \text{for } i \sim j. \tag{9}$$

Note that $\widehat{W}_{ij} = -\widehat{W}_{ji}$ if $i \sim j$ is skew-symmetric. We then seek an embedding $y$ by minimizing the sum of squared discrepancies between the embedding differences $y_j - y_i$ and these target increments $\widehat{W}_{ij}$ over all neighboring pairs:

$$\min_{y \in \mathbb{R}^n} \sum_{i,j:\, i \sim j} \left(y_j - y_i - \widehat{W}_{ij}\right)^2. \tag{10}$$

This objective function is invariant to a constant shift in $y$ (i.e., if $y$ is a solution, then $y + c\mathbf{1}$ is also a solution for any $c \in \mathbb{R}$), meaning the solution is not unique. To ensure a unique solution (up to global sign, which doesn't affect ordering), we impose a constraint, typically by centering the embedding: $\langle \mathbf{1}, y \rangle = \sum_i y_i = 0$. The optimization problem thus becomes:

$$\widehat{y} = \underset{y \in \mathbb{R}^n:\, \langle \mathbf{1}, y \rangle = 0}{\operatorname{argmin}} \sum_{i,j:\, i \sim j} \left(y_j - y_i - \widehat{W}_{ij}\right)^2. \tag{11}$$

This is a standard least-squares problem whose solution can be found by solving a linear system related to the graph Laplacian.

**Lemma 1** (Laplacian form). *Let $A$ be the adjacency matrix of the neighborhood graph (where $A_{ij} = 1$ if $i \sim j$ and $0$ otherwise), and let $L = D_A - A$ be the corresponding graph Laplacian matrix, where $D_A$ is the diagonal matrix of node degrees. Let $\widehat{\eta} \in \mathbb{R}^n$ be a vector with entries $\widehat{\eta}_i = \sum_{j=1}^n A_{ij}\widehat{W}_{ij}$. Minimizing the objective function in (11) is equivalent to*

$$\underset{y \in \mathbb{R}^n:\, \langle \mathbf{1}, y \rangle = 0}{\operatorname{argmin}} \quad y^T L y - 2y^T \widehat{\eta}. \tag{12}$$

*The unique optimal solution $\widehat{y}$ to (11) is given by $\widehat{y} = L^+ \widehat{\eta}$, where $L^+$ is the Moore-Penrose pseudo-inverse of $L$.*

### 3.4 FINAL ORDERING

Given the computed 1D embedding $\widehat{y} = (\widehat{y}_1, \ldots, \widehat{y}_n)^\intercal$, the final estimated linear order of the original data points is obtained by sorting these embedding values. We define the permutation $\phi$ of the indices $\{1, \ldots, n\}$ as:

$$\phi = \operatorname{argsort}_{j \in \{1, \ldots, n\}} \widehat{y}_j. \tag{13}$$

This permutation $\phi$ provides the sequence $(\phi(1), \phi(2), \ldots, \phi(n))$ such that $\widehat{y}_{\phi(1)} \leq \widehat{y}_{\phi(2)} \leq \cdots \leq \widehat{y}_{\phi(n)}$, representing the recovered seriation.

## 4 THEORETICAL ANALYSIS

Consider $X = \gamma(\Theta) + \varepsilon$ with $\Theta \sim f_\Theta$ independent of the noise $\varepsilon$, of variance $\sigma^2$. Let $f_{\max}$ and $f_{\min}$ be the maximum and minimum of $f_\Theta$, the density of $\Theta$. We will make the following assumptions:

**Assumption 1.** *$\gamma$ is parameterized by arc-length, so that $\gamma'(\theta)$ is well-defined and $\|\gamma'(\theta)\| = 1$ for all $\theta$. The curvature of $\gamma$ is bounded by $\kappa$.*

**Assumption 2.** *The noise is bounded so that $\{x_i\}$ lie in a tubular neighborhood $\mathcal{T}_\rho = \{x \in \mathbb{R}^d : \mathrm{dist}(x, \mathcal{M}) \leq \rho\}$ around the manifold, for some $\rho > 0$.*

**Assumption 3.** *For some $\delta \in (0, 1)$, and a sufficiently large numerical constant $C_1 > 0$, we have $(1 - \delta)r \geq \rho + 2\sqrt{d}\sigma$ and $\sqrt{d}\sigma \leq \delta r \leq \frac{1}{\kappa}$ and*

$$(\delta r)^3 \geq \frac{C_1}{f_{\min}} G_n(\sigma, \rho, \kappa) \quad where$$

$$G_n(\sigma, \rho, \kappa) := \sigma^2 + \rho^2 + \kappa^2(r^4 + \rho^4) + \sqrt{\frac{d}{n}}.$$

For $r$ of constant order, these conditions are satisfied for sufficiently large $n$ and small $\kappa$, $\rho$ and $\sigma$. In particular, note that $G_n(\sigma, \rho, \kappa) \to 0$ as $n \to \infty$ and $\kappa, \sigma, \rho \to 0$.

We also need the following assumption to prevent the near self-intersection of the curve

**Assumption 4.** *For any $\theta_0$ in the domain of $\gamma$, the set $\{\theta : \gamma(\theta) \in B(\gamma(\theta_0), 2\rho + r)\}$ is included in the interval $[\theta_0 - \frac{\pi}{2\kappa}, \theta_0 + \frac{\pi}{2\kappa}]$.*

This holds, for example, under the stronger assumption $2L\kappa \leq \pi$ where $L$ is the total length of the curve. This assumption implies that the curve does not come close to intersecting itself: For data points on $\gamma$ that are close in $\mathbb{R}^d$, their inverse image (via $\gamma^{-1}$) are also close in the domain of $\gamma$. In other words, the neighborhood of any point on the curve large enough to capture noisy points contains only "one branch" of the curve.

Let $d_{\mathrm{avg},2} = \sqrt{\frac{1}{n}\sum_i d_i^2}$ be an average degree, and $\lambda_2(L)$ the spectral gap of the Laplacian $L$ of the $r$-neighborhood graph $G_r(\mathscr{X}_n)$.

For vectors $x$ and $y$, $\tau(x, y)$ denotes their Kendall's $\tau$ coefficient, a measure of their rank similarity. We have the following result:

**Theorem 1** (Main result). *Assume $\kappa(2\rho + r) \ll 1$. Then, with probability at least $0.99$, we have*

$$1 - \tau(\widehat{y}, \theta - \bar{\theta}) \lesssim f_{\max}^{2/3}\left(\frac{d_{avg,2}}{\lambda_2(L)}\right)^{2/3}\left(K_\kappa(2\rho + r) + \frac{G_n(\sigma, \rho, \kappa) \cdot r}{f_{\min}(\delta r)^3} + \rho\right)^{2/3} + n^{-1/2}.$$

*where $K_\kappa(t) := \kappa t^2(1 + \frac{1}{4}\kappa t)$.*

See Appendix B for the proof. The bound follows from Proposition 1, which converts MSE between vectors $\theta, y \in \mathbb{R}^n$ into a Kendall tau error bound $1 - \tau(\theta, y)$. Bounding $f_{\max}$ (the maximum density of $\Theta$) is essential: without it, MSE could be small while $1 - \tau$ remains large. Intuitively, bounding $f_{\max}$ prevents $\theta_i$ from concentrating too much in a small region, which would allow large ranking changes with only small $\ell_2$ movements.

The term

$$\frac{G_n}{f_{\min}(\delta r)^3} = \frac{\sigma^2 + \rho^2 + \kappa^2(r^4 + \rho^4)}{f_{\min}(\delta r)^3} + \frac{\sqrt{d/n}}{f_{\min}(\delta r)^3}$$

captures tangent vector estimation error, where the first term is bias (approximation error) and the second is roughly variance (estimation error)—see Lemmas 4 and 5 and (17). The term $K_\kappa(2\rho + r)$ bounds the gap between $\widehat{W}_{ij}$ and its ideal value $\theta_j - \theta_i$. Even with no noise and exact tangent vectors, curvature $\kappa$ creates a discrepancy, which $K_\kappa(2\rho + r)$ upper bounds (Lemma 8). Note this vanishes as $\kappa \to 0$.

## 4.1 INTERPRETATION OF THE RESULT

To interpret the result, it helps to think of $f_{\min}$ and $f_{\max}$ as constants (the minimum and maximum of the density function of $\Theta$): the ideal case is the uniform distribution on $[0, L]$, where $f_{\min} = f_{\max} = 1/L$ and $L$ is the length of the curve, a constant. We also take $r$ to be of constant order. Then, the quantity $\frac{d_{\mathrm{avg},2}}{\lambda_2(L)}$ is expected to be of constant order (cf. Von Luxburg et al. (2014)), hence can be absorbed into the constants. So the quantity $f_{\max}^{2/3}\left(\frac{d_{\mathrm{avg},2}}{\lambda_2(L)}\right)^{2/3} \lesssim 1$ and can be ignored.

The quality of the bound is then controlled by the curvature $\kappa$, noise level $(\sigma, \rho)$ and the sample size $n$. Consider the easiest case where $\gamma(\cdot)$ is a straight line, so that $\kappa = 0$ and we have no noise ($\sigma = 0$, so we can take $\rho = 0$ as well). In this case $K_\kappa(2\rho + r) = 0$ and $G_n(\sigma, \rho, \kappa) = \sqrt{d/n}$, and the the bound simply reads

$$1 - \tau(\widehat{y}, \theta - \bar{\theta}) \lesssim \left( \frac{\sqrt{d/n} \cdot r}{f_{\min}(\delta r)^3} \right)^{2/3} + n^{-1/2} \lesssim \left( \frac{d}{n} \right)^{1/3} + \frac{1}{\sqrt{n}}$$

where the second inequality further absorbed the constants $r, \delta$ and $f_{\min}$ into $\lesssim$. This bound shows that in the noiseless linear case, we achieve consistency as long as $n \to \infty$ (and $d = o(n)$).

**Fundamental information limit.** In the presense of noise, consistency is not possible: that is, adding (isotropic) noise, it will not be possible to recover the order consistently, in the sense of driving $1 - \tau$ to zero. This is a fundamental information limit of the problem, experienced by any method, not a deficiency of our method. To see this, just consider the case where $\gamma(\theta) = \theta$, a straight line in $\mathbb{R}$. Adding noise $\varepsilon_i$ to $\theta_i \sim U[0, 1]$ swaps the order of a positive fraction of pairs of points $\{\theta_i + \varepsilon_i\}_{i=1}^n$ as $n \to \infty$. A positive curvature exacerbates the problem (a smaller amount noise can swap the order for higher curvature).

A consequence of the above discussion is that for the $\sigma > 0$ regime, all one can hope for a method is to have an error that can be made arbitrarily small in the limit of $n \to \infty$, by making $\kappa$ and $\sigma$ small enough. This is exactly what Theorem 1 shows for STAGE. Asymptotically (as $n \to \infty$), we have

$$\lim_{n \to \infty} [1 - \tau(\widehat{y}, \theta - \bar{\theta})] \lesssim \left( K_\kappa(2\rho + \sigma^2) + \rho^2 + \kappa^2(1 + \rho^4) + \rho \right)^{2/3}$$

and the RHS can be made arbitrarily small by making $\kappa$, $\sigma$ and $\rho$ small enough.

## 5 EXPERIMENTS

We evaluate the empirical performance of our proposed STAGE algorithm against several existing seriation methods. All experiments were conducted on a desktop computer equipped with an AMD Ryzen 9 7900X processor and 32 GB of RAM.

### 5.1 SYNTHETIC DATA

We first assess STAGE on synthetic datasets, comparing with established algorithms: the spectral method by Atkins et al. (1998), 1D UMAP (McInnes et al., 2018), 1D t-SNE (Van der Maaten &

Table 1: Comparison of Methods ($n = 2000$), Kendall's $\tau$ and std. err. values multiplied by 100.

| Dim. | Method | $\sigma = 0.5$ | $\sigma = 1$ | $\sigma = 1.5$ | $\sigma = 2$ |
|---|---|---|---|---|---|
| 50 | Fiedler Vector | 85.47 (4.53) | 76.91 (3.53) | 49.49 (11.23) | 33.22 (14.77) |
| | 1D UMAP | 88.6 (0.67) | 77.08 (1.89) | 64.5 (3.41) | 51.17 (3.5) |
| | 1D t-SNE | 90.51 (0.63) | 81.03 (1.43) | 71.53 (2.75) | 61.51 (3.16) |
| | Recanati | 90.05 (1.03) | 80.1 (1.35) | 71.6 (1.86) | **64.39 (2.35)** |
| | STAGE | **90.91 (0.66)** | **81.76 (1.47)** | **72.26 (2.36)** | 64.04 (2.58) |
| 100 | Fiedler Vector | 83.1 (12.9) | 83.34 (1.55) | 63.0 (13.29) | 44.9 (11.3) |
| | 1D UMAP | 90.89 (0.45) | 81.72 (1.18) | 71.14 (2.07) | 58.85 (4.81) |
| | 1D t-SNE | 92.93 (0.33) | 85.66 (0.86) | 78.28 (1.49) | 70.05 (3.35) |
| | Recanati | 92.77 (0.47) | 84.51 (0.76) | 77.9 (1.03) | 72.07 (1.39) |
| | STAGE | **93.37 (0.29)** | **86.46 (0.85)** | **79.47 (1.39)** | **72.27 (1.89)** |
| 200 | Fiedler Vector | 68.8 (22.04) | 88.7 (0.37) | 81.2 (1.1) | 66.52 (4.1) |
| | 1D UMAP | 93.29 (0.33) | 86.35 (0.46) | 78.89 (0.62) | 70.17 (1.03) |
| | 1D t-SNE | 95.01 (0.08) | 89.94 (0.26) | 84.98 (0.28) | 77.47 (4.18) |
| | Recanati | 95.17 (0.12) | 87.84 (0.21) | 83.41 (0.34) | 78.88 (0.49) |
| | STAGE | **95.47 (0.09)** | **90.77 (0.24)** | **86.12 (0.28)** | **81.18 (0.56)** |

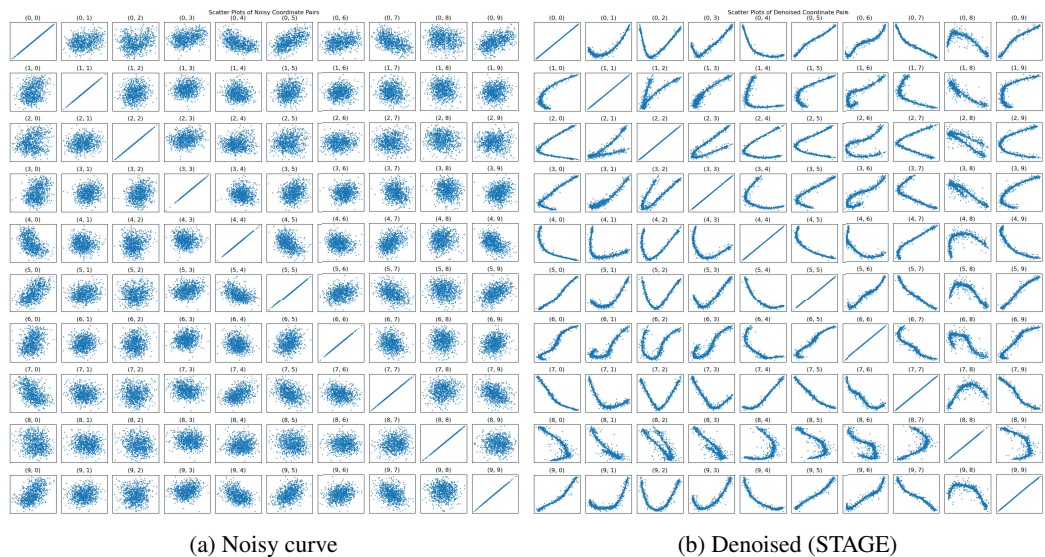

(a) Noisy curve             (b) Denoised (STAGE)

Figure 2: Pairplots of the first 10 dimensions: (a) original noisy observations, (b) denoised points produced by STAGE.

Hinton, 2008), and the method by Recanati et al. (2018). We include some experiments on curves in low dimensional simulation data in Section E.1 in the appendix, but here we focus on its performance in higher dimensions. Performance is quantified using Kendall's $\tau$ rank correlation coefficient between the true and recovered orderings.

For these experiments we utilize curves based on random Fourier series, letting the $i$-th dimension of $\gamma$ take the form

$$\gamma_i(t) = \sum_{j=1}^{J} a_{ij} \sin(2\pi jt) + b_{ij} \cos(2\pi jt)$$

where $a_{ij}, b_{ij} \sim \mathcal{N}(0, j^{-\alpha})$ for some choice of $\alpha$ and $K$. By increasing $K$ or decreasing $\alpha$ we can increase the curvature of the resulting manifold, giving us a modular method for creating curves in any dimension. These curves as described are closed, i.e. they begin and end at the same point with $t = 0$ and $t = 1$. To account for this we take the path on only a quarter of its span, between $t = 0$ to $t = 0.25$, to give us an open curve that is more suitable for the problem we aim to solve. We plot the first 3 dimensions of an example curve in Figure 5 in the appendix. Lastly, we modify our algorithm so that neighborhoods are built through $k$-NN with $k = 50$ rather than through a ball, as finding sufficiently many points in a ball can be difficult in high dimension. We test our method on curves generated with $J = 10$ and $\alpha = 2.3$, and over dimensions $d \in \{50, 100, 200\}$, with noise levels $\sigma \in \{.5, 1, 1.5, 2\}$ and sample sizes $n \in \{500, 1000, 2000\}$, averaging over 10 runs with each combination of settings. In Table 1, we see that STAGE consistently beats the other methods in terms of $\tau$ for n = 2000 (the full results are displayed in Table 4 in the appendix with similar results), noting that STAGE takes significantly less time than the competitors to run, and that while our choice of neighborhood size is static, typically STAGE improves when neighborhood size increases with sample size, which we demonstrate experimentally in the appendix versus t-SNE. Figure 2 shows the results of applying STAGE to the first 10 dimensions of a random Fourier curve, demonstrating that STAGE not only recovers the correct ordering but also denoises the data. For a full comparison with the true curve, see Figure 6 in the appendix.

## 5.2 REAL DATA: SINGLE-CELL RNA-SEQ PSEUDOTIME ORDERING

To assess STAGE's applicability to real-world biological data, we applied it to the task of pseudo-time ordering in single-cell RNA-Seq (scRNA-Seq) experiments. The goal is to infer a temporal progression of cells based on their gene expression profiles, reflecting processes such as development or differentiation. We utilized three different datasets to assess our method. The first is a publicly

|                    | Fiedler | 1D UMAP | 1D t-SNE | Recanati | STAGE |
|--------------------|---------|---------|----------|----------|-------|
| Saelens            | 0.695   | 0.634   | **0.808**| 0.800    | 0.786 |
| Packer Seam Cells  | 0.817   | 0.589   | 0.889    | 0.792    | **0.926** |
| Packer Hypodermis  | 0.743   | 0.472   | 0.801    | 0.712    | **0.822** |

Table 2: Performance of seriation methods based on Kendall's $\tau$ on the three RNA-Seq datasets

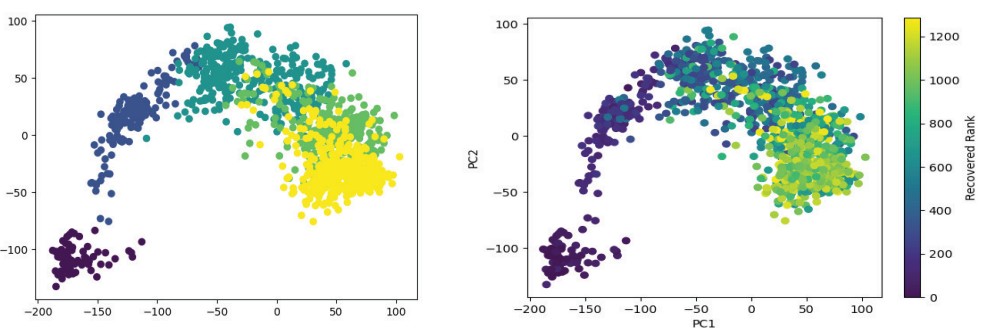

Figure 3: (Left) RNA-seq data reduced to 2 PCs and colored by actual stage, (Right) RNA-seq data reduced to 2 PCs and colored by recovered order

available dataset of human pre-implantation embryo development from Petropoulos et al. (2016), as processed and benchmarked in Saelens et al. (2019). After some additional processing and filtering, this dataset comprises 1289 cells and 6058 gene expression measurements, annotated with discrete developmental stages rather than a continuous ground-truth pseudotime. The other two datasets are subsets of the data collected by Packer et al. (2019), tracking the development of *C. elegans* cells over time. In particular, we consider the development of seam cells and hypodermis cells over time. The former contains 2766 cells and 14870 genes, and the latter 7746 cells and 16379 genes.

For each data set, we applied Principal Component Analysis (PCA) to the gene expression data for dimensionality reduction and visualization (Figure 3, left). We then ran STAGE on the first twenty principal components (PCs) to infer an ordering. We present the results on the three datasets in Table 2, where we can see that we perform slightly worse than t-SNE and Recanati on the Saelens dataset, but better on the two Packer datasets. We also include a visualization of the Saelens data in 2 PCs in Figure 3 which demonstrates that the STAGE embedding, when visualized against the known developmental stages (Figure 3, right), generally aligns with the expected biological progression. While some mixing of adjacent stages is observed, which is consistent with the inherent overlap and stochasticity in developmental processes and the discrete nature of the ground-truth labels, STAGE successfully captures the broad temporal trajectory.

An important consideration for such results is the selection of $r$. Briefly, we have found that it is important to select a radius such that the resulting neighborhood graph is connected. In practice, a good radius may be somewhere between 1 and 2 times the minimum radius necessary to generate this connected neighborhood graph. We include further investigation of this in the appendix.

## 6 DISCUSSION AND LIMITATIONS

Our STAGE algorithm is tailored for linear seriation on 1D manifolds. While local PCA is applicable in higher dimensions, generalizing STAGE for $k$-dimensional manifold learning ($k > 1$) faces significant challenges.

**Topological Constraint: Orientability.** A key simplification for 1D manifolds is their inherent orientability, allowing for a globally consistent direction. Our spectral sign alignment (Section 3.2) leverages this to establish such a consistency. For non-orientable manifolds of dimension $k \geq 2$

(e.g., a 2D Möbius strip), a globally consistent orientation of tangent spaces does not exist. Thus, our current alignment strategy, which finds a single consistent orientation, would not directly apply. Determining manifold orientability from data is a non-trivial precursor task.

**Methodological Generalization.** Extending STAGE to $k > 1$ dimensions, even for orientable manifolds, presents further difficulties:

1. **Orientation Alignment:** Aligning $k$-dimensional tangent bases (frames) requires finding optimal orthogonal transformations (e.g., elements of $O(k)$ xbetween neighbors. This is a more complex synchronization problem than the $\mathbb{Z}_2$ (sign) synchronization solved by Equation (7), demanding more sophisticated optimization techniques.

2. **Global Embedding:** The least-squares embedding (Section 3.3) would need to be reformulated to produce a $k$-dimensional output $y_i \in \mathbb{R}^k$, with the target increments $\widehat{W}_{ij}$ also becoming $k$-dimensional vectors.

**Future Work Considerations.** Future work might explore adapting STAGE for $k$-dimensional manifold learning, assuming orientability. This would involve developing methods for $k$-frame synchronization and a corresponding $k$-dimensional geometric embedding. However, the present work focuses on the specific advantages STAGE offers for the 1D linear seriation problem.

## 7 CONCLUSION

We presented STAGE (Spectral Tangent Alignment and Geometric Embedding), a geometric pipeline for linear seriation that works directly on noisy point clouds sampled from a 1D manifold. STAGE combines three stages: local tangent estimation via neighborhood PCA, spectral sign alignment for global orientation consistency, and a Laplacian least-squares embedding driven by oriented local projections. This preserves coordinate-level geometric cues that are often lost in similarity-matrix approaches, while remaining simple and efficient.

Our analysis provides finite-sample guarantees linking curvature, noise, neighborhood scale, and graph connectivity to rank recovery, yielding bounds on Kendall's $\tau$ that approach 1 under benign geometry and noise. Empirically, STAGE delivers accurate and fast order recovery on synthetic high-dimensional curves and real single-cell RNA-seq pseudotime data, often matching or surpassing strong baselines, and producing denoised embeddings that reflect manifold structure.

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

## A ON INFORMATION LOSS IN SIMILARITY-BASED APPROACHES

Here we clarify in what sense working solely with a global similarity (or distance) matrix can discard geometric information that our method exploits, and illustrate this with a synthetic experiment.

Using the notation of Section 3.3, we note that $\widehat{W}_{ij}$ is the *signed distance along a tangent line*, and ideally given by $v_{ij}^T(x_i - x_j)$ where $v_{ij}$ is a good estimate of the tangent direction near $x_i$ and $x_j$. Our approach then solves

$$\widehat{y} = \arg \min_{y \in \mathbb{R}^n: \langle \mathbf{1}, y \rangle = 0} \sum_{i,j: i \sim j} \left( y_j - y_i - \widehat{W}_{ij} \right)^2.$$

On the other hand, a method that the relies on, say, a distance-based similarity (such as the Gaussian kernel), ideally tries to reconstructs a 1-D representation of the *squared distance*:

$$\widehat{y} = \arg \min_{y \in \mathbb{R}^n: \langle \mathbf{1}, y \rangle = 0} \sum_{i,j: i \sim j} \left( (y_j - y_i)^2 - \|x_i - x_j\|^2 \right)^2.$$

Consider the case where $x_i$ lie on a line and noise is orthogonal to the line: for example, $x_i = (\theta_i, 0) + (0, \varepsilon_i)$. Assuming we obtain a good estimate of $v_{ij}$ (correctly aligned), we get $\widehat{W}_{ij} \approx \theta_j - \theta_i$ since $v_{ij}$ nearly eliminates the noise. On the other hand, distance-based approaches observe $\|x_i - x_j\|^2 = (\theta_j - \theta_i)^2 + (\varepsilon_j - \varepsilon_i)^2$. For sufficiently large noise, the signal can be completely washed out.

Even if one can reconstruct the distances along the tangent line from the similarity matrix (e.g., Recanati et al. (2017) attempts this by first embedding the points via Laplacian embedding), they would still optimize something like:

$$\widehat{y} = \arg \min_{y \in \mathbb{R}^n: \langle \mathbf{1}, y \rangle = 0} \sum_{i,j: i \sim j} \left( (y_j - y_i)^2 - \widehat{W}_{ij}^2 \right)^2.$$

In general, $W_{ij} = \theta_i - \theta_j + \delta_{ij}$. Assuming $\delta_{ij}$ is nearly zero-mean, we have $\mathbb{E}[\widehat{W}_{ij}^2] \approx (\theta_i - \theta_j)^2 + \mathbb{E}[\delta_{ij}^2]$ for this approach versus $\mathbb{E}[\widehat{W}_{ij}] \approx (\theta_i - \theta_j)$ for our approach since $\mathbb{E}[\delta_{ij}] \approx 0$. Thus the squared distance approach suffers from the extra noise term $\mathbb{E}[\delta_{ij}^2]$.

It is worth noting that if the similarity matrix is already built from squared distances, the information is already lost in the anisotropic example and embedding back will not help.

**Synthetic experiment.** We illustrate this gap quantitatively. Consider equispaced points $\theta_i$ on $[0, 1]$ and let $x_i = (\theta_i, 0) + (\varepsilon_{i,\|}, \varepsilon_{i,\perp})$, with $\varepsilon_{i,\|} \sim \mathcal{N}(0, \sigma_\|^2)$ and $\varepsilon_{i,\perp} \sim \mathcal{N}(0, \sigma_\perp^2)$ independent. We fix $\sigma_\| = 10^{-3}$ and vary $\sigma_\perp$ over a grid $\sigma_\perp \in \{0.10, 0.13, \ldots, 0.35\}$. For Recanati's method Recanati et al. (2018), we construct a Gaussian similarity matrix with bandwidth $\sigma = (\sigma_\|^2 + \sigma_\perp^2)^{1/2}$, while STAGE operates directly on the coordinates via local tangent estimation as in the main text. Figure 4 reports the Kendall's $\tau$ between the recovered order and the ground-truth order, averaged over 5 repetitions.

We observe that Recanati's $\tau$ degrades faster than STAGE. Only when $\sigma_\perp$ is so large that the geometry is essentially unrecoverable for both methods do their performances converge.

## B DETAILED PROOF OF THEOREM 1

**Preliminary results.** We start with a couple of lemmas. The first establishes the concentration of local emprical covariance matrix:

**Lemma 2** (Concentration). *On the event $|\{j : j \sim i\}| = d_i$, With probability at least $1 - 2e^{-\delta}$,*

$$\|\widehat{\Sigma}_i - \Sigma(x_i, h)\| \lesssim \sqrt{\frac{d + \delta}{d_i}}.$$

*Proof.* The result follows from Exercise 4.7.3 of Vershynin (2018). $\square$

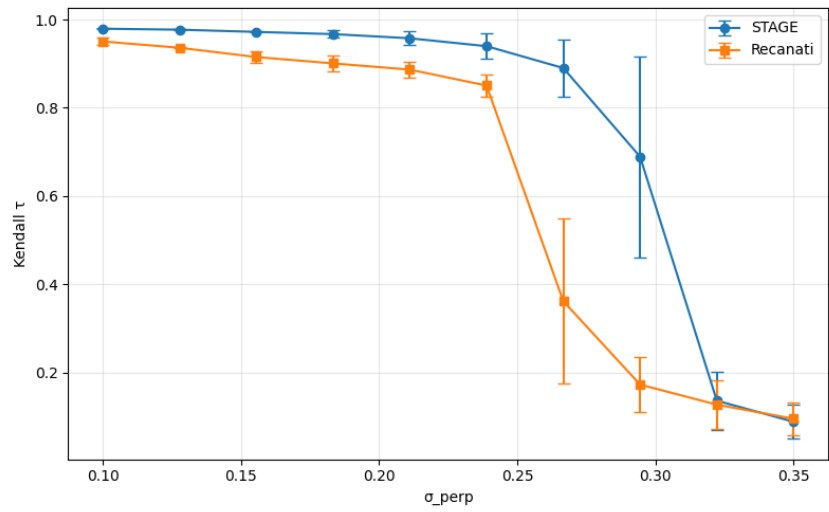

Figure 4: Kendall's $\tau$ between the recovered order and the ground-truth order, averaged over 5 repetitions, as a function of $\sigma_\perp$.

The next result is well-known from differential geometry:

**Lemma 3** (Chord length inequality). *Let $\gamma : I \to \mathbb{R}^d$ be a smooth curve parametrized by arc length $s$, defined on an interval $I$. Suppose the curvature $\kappa(s) = \|\gamma''(s)\|$ is bounded above by $\kappa > 0$. For any two parameter values $\theta_0, \theta \in I$ such that $|\theta - \theta_0| \leq \pi/\kappa$, the following inequalities holds:*

$$\frac{2}{\kappa} \sin\left(\frac{\kappa}{2}|\theta - \theta_0|\right) \leq \|\gamma(\theta) - \gamma(\theta_0)\|,$$

$$\frac{1}{\kappa} \sin(\kappa|\theta - \theta_0|) \leq |\gamma'(\theta_0)^\intercal(\gamma(\theta) - \gamma(\theta_0))|.$$

We frequently combine the bounds with the elementary inequality $\sin(x) \geq \frac{2}{\pi}x$ for $0 \leq x \leq \pi/2$. In particular, assumuing $\kappa|\theta - \theta_0| \leq \pi/2$, we have

$$\frac{2}{\pi}|\theta - \theta_0| \leq \|\gamma(\theta) - \gamma(\theta_0)\| \tag{14}$$

$$\frac{2}{\pi}|\theta - \theta_0| \leq |\gamma'(\theta_0)^\intercal(\gamma(\theta) - \gamma(\theta_0))|. \tag{15}$$

Let us now introduce some notation used in the spectral analysis of the local covariance matrices. For two (unit) vectors $u, v \in \mathbb{R}^d$, we write

$$d_\pm(u, v) := \min_{s \in \{\pm 1\}} \|u - sv\|$$

Note that $|\langle u, v \rangle| = 1 - \frac{1}{2}d_\pm(u, v)$.

For any point $\theta \in [0, 1]$, let $v(\theta) := \gamma'(\theta)$ be the tangent vector to the curve at $\theta$.

For any point $x_0 \in \mathbb{R}^d$, other than the two endpoints, let $\pi(x_0)$ be a projection of $x_0$ onto the curve (a nearest point to $x_0$ on $\mathcal{M}$). Letting $\theta_0 = \gamma^{-1}(\pi(x_0))$, by smoothness of the curve, $v(\theta_0)$ will be orthogonal to $x_0 - \pi(x_0)$. Let $\rho_0 = \|x_0 - \pi(x_0)\|$ be the distance of $x_0$ to the curve.

We define

$$\Sigma(x_0, r) := \mathbb{E}[(x - x_0)(x - x_0)^T \mid x \in B(x_0, r)]$$

as the population version of (4), where $x = \gamma(\theta) + \varepsilon$. Here, $x$ and $\theta$ are random variables, while $x_0$ and $\theta_0$ are fixed.

Let $\sigma_1(x_0, r)$ be the top eigenvalue of $\Sigma(x_0, r)$. With $x_0$ and $\theta_0$ related as described above, we let

$$\Sigma^*(x_0, r) = \sigma_1(x_0, r)\, v(\theta_0)v(\theta_0)^\intercal.$$

The next result provides a lower bound on the leading eigenvalue of $\Sigma(x_0, r)$, namely $\sigma_1(x_0, r)$.

**Lemma 4** (Top eigenvalue). *Assume that $(1 - \delta)r \geq \rho + 2\sqrt{d}\sigma$ for some $\delta \in (0, 1)$. Then,*

$$\mathbb{E}[(\gamma'(\theta_0)^\mathsf{T}(x - x_0))^2 \mathbf{1}_{B_{x_0}(r)}(x)] \geq C_0 f_{\min} \cdot u \cdot ((d\sigma^2) \vee u^2)$$

*where $u = \frac{1}{\kappa} \wedge (\delta r)$ for some numerical constant $C_0 > 0$. In particular, assuming $\sqrt{d}\sigma \leq \delta r \leq \frac{1}{\kappa}$, the lower boudns simplifies to $C_0 f_{\min}(\delta r)^3$.*

*Proof.* Recall that $\pi(x_0) = \gamma(\theta_0)$. Since $x_0 - \gamma(\theta_0)$ is perpendicular to $\gamma'(\theta_0)$, we have $\gamma'(\theta_0)^\mathsf{T}(x - x_0) = \gamma'(\theta_0)^\mathsf{T}(x - \gamma(\theta_0))$. Thus, we want to control

$$\mathbb{E}[(\gamma'(\theta_0)^\mathsf{T}(x - \gamma(\theta_0)))^2 \mathbf{1}_{B_{x_0}(r)}(x)] = \mathbb{E}[(\gamma'(\theta_0)^\mathsf{T}(\gamma(\theta) + \varepsilon - \gamma(\theta_0)))^2 \mathbf{1}_{B_{x_0}(r)}(x)].$$

Recall that $\theta$ and $\varepsilon$ are random variables, while $\theta_0$ and $x_0$ are fixed. Let us define the event

$$\mathcal{A} := \left\{|\theta - \theta_0| \leq \min\left(\frac{\pi}{2\kappa}, r - \rho - 2\sqrt{d}\sigma\right)\right\}, \quad \mathcal{E} := \{\|\varepsilon\| \leq 2\sqrt{d}\sigma\}.$$

By Lemma 3, in particular (15), on $\mathcal{A}$, we have

$$|\gamma'(\theta_0)^\mathsf{T}(\gamma(\theta) - \gamma(\theta_0))| \geq \frac{2}{\pi}|\theta - \theta_0|.$$

Recall that $\gamma(\theta_0) = \pi(x_0)$, hence

$$x - x_0 = (\gamma(\theta) - \gamma(\theta_0)) + (\pi(x_0) - x_0) - \varepsilon. \tag{16}$$

We observe that

$$\|\varepsilon\| \leq 2\sqrt{d}\sigma \quad \text{and} \quad |\theta - \theta_0| \leq r - \rho - 2\sqrt{d}\sigma \implies \|x - x_0\| \leq r,$$

which follows from $\|\gamma(\theta) - \gamma(\theta_0)\| \leq |\theta - \theta_0|$ and $\|\pi(x_0) - x_0\| \leq \rho$ and the triangle inequality. This implies that $\mathcal{A} \cap \mathcal{E} \subseteq B_{x_0}(r)$.

Moreover, on $\mathcal{E}$, we have $|\gamma'(\theta_0)^\mathsf{T}\varepsilon| \leq 2\sqrt{d}\sigma$. Thus, on $\mathcal{A} \cap \mathcal{E}$, we have, by triangle inequality,

$$|\gamma'(\theta_0)^\mathsf{T}(\gamma(\theta) + \varepsilon - \gamma(\theta_0))| \geq \left|\frac{2}{\pi}|\theta - \theta_0| - 2\sqrt{d}\sigma\right|.$$

Squaring both sides and noting that $\mathbf{1}_{\mathcal{A} \cap \mathcal{E}} \leq \mathbf{1}_{B_{x_0}(r)}(x)$, we can derive the lower bound

$$F := \mathbb{E}[(\gamma'(\theta_0)^\mathsf{T}(\gamma(\theta) + \varepsilon - \gamma(\theta_0)))^2 \mathbf{1}_{B_{x_0}(r)}(x)] \geq \mathbb{E}\left[\left(\frac{2}{\pi}|\theta - \theta_0| - 2\sqrt{d}\sigma\right)^2 \mathbf{1}_{\mathcal{A} \cap \mathcal{E}}\right].$$

$$= \mathbb{E}\left[\left(\frac{2}{\pi}|\theta - \theta_0| - 2\sqrt{d}\sigma\right)^2 \mathbf{1}_{\mathcal{A}}\right]\mathbb{P}(\mathcal{E})$$

$\mathbb{P}(\mathcal{E})$ is lower bounded by an absolute constant. Let $b := \min\{\frac{\pi}{2\kappa}, r - \rho - 2\sqrt{d}\sigma\}$, so that $\mathcal{A} = \{|\theta - \theta_0| \leq b\}$. Then,

$$F_1 := \mathbb{E}\left[\left(\frac{2}{\pi}|\theta - \theta_0| - 2\sqrt{d}\sigma\right)^2 \mathbf{1}_{\mathcal{A}}\right] = \int f_\Theta(\theta)\left(\frac{2}{\pi}|\theta - \theta_0| - 2\sqrt{d}\sigma\right)^2 \mathbf{1}_{\{|\theta - \theta_0| \leq b\}}d\theta$$

$$\geq 2f_{\min}\int_0^b \left(\frac{2}{\pi}t - 2\sqrt{d}\sigma\right)^2 dt$$

$$\geq f_{\min}\frac{\pi}{3}\left(\left(\frac{2}{\pi}b - 2\sqrt{d}\sigma\right)^3 + (2\sqrt{d}\sigma)^3\right).$$

Let $\frac{2}{\pi}b = x$ and $2\sqrt{d}\sigma = a$. We have $(x - a)^3 + a^3 = xh(x)$ where $h(x)$ is the quadratic function $x^2 - 3ax + 3a^2$, which has a global minimum of $\frac{3}{4}a^2$. We also note that $h(x) - \frac{x^2}{4} = \frac{3}{4}(x - 2a)^2 \geq 0$. Thus, we have $h(x) \geq \frac{1}{4}\max\{3a^2, x^2\}$.

By assumption $(1 - \delta)r \geq \rho + 2\sqrt{d}\sigma$, we have $b \geq \min\{\frac{\pi}{2\kappa}, \delta r\} \geq \frac{1}{\kappa} \wedge (\delta r) =: u$. We obtain

$$F_1 \geq f_{\min}\frac{\pi}{3}x \cdot \frac{1}{4}\max\{3a^2, x^2\}$$

$$= f_{\min} \cdot \frac{b}{6}\max\{3(2\sqrt{d}\sigma)^2, (2b/\pi)^2\} \gtrsim f_{\min}u((d\sigma^2) \vee u^2).$$

Combined with $\mathbb{P}(\mathcal{E}) \gtrsim 1$, the proof is complete. $\square$

The next lemma provides an upper bound on the *second* largest eigenvalue of $\Sigma(x_0, r)$:

**Lemma 5** (Second largest eigenvalue). *Under Assumption 4, for any $v$ perpendicular to $\gamma'(\theta_0)$,*

$$\mathbb{E}[(v^\mathsf{T}(x - x_0))^2 \mathbf{1}_{B_{x_0}(r)}(x)] \leq C_0'\Big(\sigma^2 + \rho^2 + \kappa^2(r^4 + \rho^4)\Big)$$

*for some numerical constant $C_0' > 0$.*

*Proof.* Given the above setting, we want to find an upper bound for

$$\mathbb{E}[(v^\mathsf{T}(x - x_0))^2 \mathbf{1}_{B_{x_0}(r)}(x)] = \mathbb{E}[(v^\mathsf{T}((x_0 - \gamma(\theta_0)) + (\gamma(\theta_0) - \gamma(\theta)) + \varepsilon))^2 \mathbf{1}_{B_{x_0}(r)}(x)].$$

for arbitrary unit $v$ perpendicular to $\gamma(\theta_0)$. We can use the inequality $(a + b + c)^2 \leq 3(a^2 + b^2 + c^2)$, so we analyze term by term. We have

$$\mathbb{E}[(v^\mathsf{T}(x_0 - \gamma(\theta_0))^2 \mathbf{1}_{B_{x_0}(r)}(x)] \leq \rho^2$$

and

$$\mathbb{E}[(v^\mathsf{T}\varepsilon)^2 \mathbf{1}_{B_{x_0}(r)}(x)] \leq \sigma^2.$$

It remains to consider the term $\mathbb{E}[(v^\mathsf{T}(\gamma(\theta) - \gamma(\theta_0))^2 \mathbf{1}_{B_{x_0}(r)}(x)]$.

From (16), we note that $\|x_0 - x\| \leq r$ implies

$$\|\gamma(\theta) - \gamma(\theta_0)\| \leq r + \rho + \|\varepsilon\| \leq r + 2\rho$$

using $\|\varepsilon\| \leq \rho$ and $\|\pi(x_0) - x_0\| \leq \rho$. Assumption 4 then implies $|\theta - \theta_0| \leq \frac{\pi}{2\kappa}$.

Using a second order Taylor expansion of $\gamma(\theta)$ around $\theta_0$, and the fact that $v^\mathsf{T}\gamma'(\theta_0) = 0$ and $\|\gamma''(\cdot)\| \leq \kappa$, we have

$$\mathbb{E}[(v^\mathsf{T}(\gamma(\theta) - \gamma(\theta_0))^2 \mathbf{1}_{B_{x_0}(r)}(x)] \leq \frac{\kappa^2}{4}\mathbb{E}\left[(\theta - \theta_0)^4 \mathbf{1}_{B_{x_0}(r)}(x)\right]$$

$$\leq \frac{\kappa^2}{4}\frac{\pi^4}{16}\mathbb{E}\left[\|\gamma(\theta) - \gamma(\theta_0)\|^4 \mathbf{1}_{B_{x_0}(r)}(x)\right]$$

where the second inequaly uses Lemma 3, in particular (14), and the fact that for $x \in B_{x_0}(r)$, the condition $|\theta - \theta_0| \leq \frac{\pi}{2\kappa}$ holds.

Using the fact that $\|\gamma(\theta) - \gamma(\theta_0)\|^4 \mathbf{1}_{B_{x_0}(r)}(x) \leq (r + 2\rho)^4 \lesssim (r^4 + \rho^4)$ finishes the proof. $\square$

The following proposition is key in converting the bound on MSE to the bound on Kendall's tau:

**Proposition 1** (From MSE to Kendall's tau). *Let $F$ be a distribution on a compact interval $[a, b] \subset \mathbb{R}$ with density $f$ satisfying $f(x) \leq f_{\max} < \infty$. Let $\theta = (\theta_1, \ldots, \theta_n)$ where $\theta_i$ are i.i.d. draws from $F$. Consider another vector $y = (y_i) \in \mathbb{R}^n$ and let $\varepsilon_n^2 = \frac{1}{n}\sum_{i=1}^n (y_i - \theta_i)^2$. Then, with probability at least $1 - \delta$,*

$$1 - \tau(y, \theta) \leq 16 \cdot 2^{-1/3}(\varepsilon_n f_{\max})^{2/3} + C\sqrt{\log(1/\delta)}\, n^{-1/2}$$

*where $C > 0$ is a universal constant.*

**Main argument.** We are now ready to prove the theorem. Fix $x_i$. We apply Lemmas 4 and 5 with $x_0 = x_i$ and $\theta_0 = \gamma^{-1}(\pi(x_i)) =: \theta_i^*$. We also let $x_i^* := \pi(x_i)$ and $v_i^* = v(\theta_i^*)$.

By Assumption 3 and Lemma 4, we have $\sigma(x_i, r) \geq C_0 f_{\min}(\delta r)^3$, and by Lemma 5,

$$\|\Sigma(x_i, r) - \Sigma^*(x_i, r)\| \lesssim \sigma^2 + \rho^2 + \kappa^2(r^4 + \rho^4) =: D_1.$$

By Lemma 2, and the fact that w.h.p. $d_i \gtrsim n$, we have w.h.p,

$$\|\widehat{\Sigma}_i - \Sigma^*(x_i, r)\| \lesssim D_1 + \sqrt{\frac{d}{n}} = G_n(\sigma, \rho, \kappa),$$

where $G_n(\sigma, \rho, \kappa)$ is defined in Assumption 3. Note that $\Sigma^*(x_i, r)$ is a rank-one matrix with leading eigenvector $v_i^* = v(\theta_i^*)$ and eigengap $\sigma_1(x_i, r) - 0$. Applying Davis–Kahan theorem, for some numerical constant $C_2 > 0$, we have

$$d_{\pm}(\widehat{v}_i, v_i^*) \lesssim \frac{\|\widehat{\Sigma}_i - \Sigma^*(x_i, r)\|}{\sigma_1(x_i, r)} \leq C_2 \frac{G_n(\sigma, \rho, \kappa)}{f_{\min}(\delta r)^3} \leq \frac{C_2}{C_1} < \frac{1}{4} \tag{17}$$

where the two final inequalities follow by applying Assumption 3 and taking $C_1$ large enough in that assumption.

Let $s_i^* \in \{\pm\}$ be the optimal sign between $\widehat{v}_i$ and $v_i^*$, that is,

$$\|s_i^* \widehat{v}_i - v_i^*\| = d_{\pm}(\widehat{v}_i, v_i^*).$$

A curvature argument gives:

**Lemma 6.** $\langle v_j^*, v_i^* \rangle \geq \frac{1}{2}$ for $i \sim j$.

Combining (17) and the preceding lemma, and applications of triangle inequality, we have $\langle s_i^* \widehat{v}_i, s_j^* \widehat{v}_j \rangle > 0$ for all $i, j$. Now we have:

**Lemma 7** (Global sign consistency). *Let $\widehat{s} = \text{sign}(\widehat{u})$ where $\widehat{u}$ is the solution of* (7). *Then, $\widehat{s} = s^*$.*

*Proof.* Consider the change of variable $\xi = (\xi_i)$ where $\xi_i = s_i^* s_i$ where $s = (s_i)$ is the optimization variable in (7). Note also that $s_i = s_i^* \xi_i$ since $s_i^* \in \{\pm 1\}$. This leads to the optimization $\widehat{\xi} = \text{argmax}_{\|\xi\|^2 = n} \xi^T W \xi$ where $W_{ij} = s_i^* s_j^* \widehat{W}_{ij} = \langle s_i^* \widehat{v}_i, s_j^* \widehat{v}_j \rangle \cdot 1\{i \sim j\}$. By the above argument $W$ is a nonnegative matrix. Recall that we assumed $G_r(\mathscr{X}_n)$ is regular, and in particular connected. Thus $W$ is a irreducible. By Perron–Frobenius theorem, its leading eigenvector has elements that are all positive, that is, $\text{sign}(\widehat{\xi}_i) = 1$ for all $i \in [n]$. Since $\widehat{u}_i = s_i^* \widehat{\xi}_i$, it follows that $\text{sign}(\widehat{u}_i) = s_i^*$. The proof is complete. $\square$

Thus, we now have

$$\|\widehat{s}_i \widehat{v}_i - v_i^*\| = d_{\pm}(\widehat{v}_i, v_i^*) \leq C_2 \frac{G_n(\sigma, \rho, \kappa)}{f_{\min}(\delta r)^3} \quad \forall i \in [n]. \tag{18}$$

Recall that $\theta_i^* := \gamma^{-1}(x_i^*)$ and $v_i^* = \gamma'(\theta_i^*)$. Recall also that the original $\theta_i = \gamma^{-1}(x_i)$. Let us define:

$$W_{ij}^* = \frac{1}{2}(v_i^* + v_j^*)^T (x_j^* - x_i^*) \cdot 1\{i \sim j\}. \tag{19}$$

**Lemma 8.** *Let $h_* = \max_{(i,j):i \sim j} |\theta_i^* - \theta_j^*|$, and write $K_\kappa(t) := \kappa t^2 (1 + \frac{1}{4}\kappa t)$. Then,*

$$\max_{(i,j):i \sim j} |W_{ij}^* - (\theta_j^* - \theta_i^*)| \leq K_\kappa(h_*).$$

Recall that $\gamma(\theta_i^*) = \pi(x_i) =: x_i^*$ for all $i$. To control $h_*$, we first note that, for $i \sim j$,

$$\|\gamma(\theta_i^*) - \gamma(\theta_j^*)\| = \|\pi(x_i) - \pi(x_j)\|$$
$$\leq \|\pi(x_i) - x_i\| + \|x_i - x_j\| + \|x_j - \pi(x_j)\| \leq \rho + r + \rho.$$

By Assumption 4, we then have $|\theta_i^* - \theta_j^*| \leq \frac{\pi}{2\kappa}$. Hence, for $i \sim j$, we can apply the chord length inequality (Lemma 3), namely (14), to obtain

$$|\theta_i^* - \theta_j^*| \leq \frac{\pi}{2}\|\pi(x_i) - \pi(x_j)\| \lesssim 2\rho + r.$$

Hence, we can take $h_* \lesssim 2\rho + r$ in Lemma (8). By a similar argument, we have

$$\|\gamma(\theta_i^*) - \gamma(\theta_i)\| \leq \|\pi(x_i) - x_i\| + \|x_i - \gamma(\theta_i)\|$$
$$\leq \rho + \|\varepsilon_i\| \leq 2\rho$$

By Assumption 4, we then have $|\theta_i^* - \theta_i| \leq \frac{\pi}{2\kappa}$. We can apply the chord length inequality (Lemma 3), namely (14), to obtain

$$|\theta_i^* - \theta_i| \leq \frac{\pi}{2} \|\gamma(\theta_i^*) - \gamma(\theta_i)\| \leq \pi\rho,$$

hence $|(\theta_i^* - \theta_j^*) - (\theta_i - \theta_i)| \leq 2\pi\rho \lesssim \rho$. Combining these results we have

$$\max_{(i,j):i\sim j} |W_{ij}^* - (\theta_j - \theta_i)| \lesssim K_\kappa(2\rho + r) + \rho.$$

Next, we note that, for $i \sim j$

$$
\begin{aligned}
2|\widehat{W}_{ij} - W_{ij}^*| &\leq \left\| (\widehat{s}_i \widehat{v}_i) + \widehat{s}_j \widehat{v}_j - (v_i^* + v_j^*) \right\| \cdot \|x_j - x_i\| \\
&\qquad + \|v_i^* + v_j^*\| \cdot \left\| (x_j - x_i) + (x_j^* - x_i^*) \right\| \\
&\leq 2C_2 \frac{G_n(\sigma, \rho, \kappa)}{f_{\min}(\delta r)^3} \cdot r + 2 \cdot 2\rho
\end{aligned}
$$

using (18) twice for the first term (via triangle inequality), and $\|v_i^*\| = \|v_j^*\| = 1$ for the second term. Putting the pieces together

$$|\widehat{W}_{ij} - (\theta_j - \theta_i)| \lesssim K_\kappa(2\rho + r) + \frac{G_n(\sigma, \rho, \kappa) \cdot r}{f_{\min}(\delta r)^3} + \rho, \quad i \sim j. \tag{20}$$

Now, we can use the following perturbation result:

**Lemma 9.** *Assume $\widehat{W}_{ij} = (\theta_j - \theta_i) + \Delta_{ij}$ for $i \sim j$ and let $\widehat{y}$ be the unique solution of (11). Then, $\widehat{y} = \theta - \bar{\theta} + L^+\eta$ where $\eta_i = \sum_j A_{ij}\Delta_{ij}$. Moreover, with probability at least 0.99,*

$$1 - \tau(\widehat{y}, \theta - \bar{\theta}) \leq 16 f_{\max}^{2/3} \left( \frac{d_{avg,2}}{\lambda_2(L)} \right)^{2/3} \|\Delta\|_\infty^{2/3} + C\sqrt{\log(100)}\, n^{-1/2}$$

*where $\|\Delta\|_\infty = \max_{i,j} |\Delta_{ij}|$ and $d_{avg,2} = \sqrt{\frac{1}{n}\sum_i d_i^2}$ is an average degree of the graph.*

Applying this result with $\|\Delta\|_\infty \lesssim$ RHS of (20), finishes the proof.

## C  REMAINING PROOFS

### C.1  PROOF OF LEMMA 1

Let $J(y) = \sum_{i,j:\, i\sim j} \left(y_j - y_i - \widehat{W}_{ij}\right)^2$ be the objective function in (11). Since $i \sim j$ implies $A_{ij} = 1$, we can write this sum over all pairs $(i,j)$ using $A_{ij}$:

$$
\begin{aligned}
J(y) &= \sum_{i=1}^n \sum_{j=1}^n A_{ij} \left(y_j - y_i - \widehat{W}_{ij}\right)^2 \\
&= \sum_{i,j} A_{ij} \left( (y_j - y_i)^2 - 2(y_j - y_i)\widehat{W}_{ij} + \widehat{W}_{ij}^2 \right) \\
&= \sum_{i,j} A_{ij}(y_j - y_i)^2 - 2\sum_{i,j} A_{ij}(y_j - y_i)\widehat{W}_{ij} + \sum_{i,j} A_{ij}\widehat{W}_{ij}^2.
\end{aligned}
$$

The first term is a standard quadratic form for the graph Laplacian: $\sum_{i,j} A_{ij}(y_j - y_i)^2 = 2y^T L y$. (This follows from $\sum_{i,j} A_{ij}(y_j^2 - 2y_i y_j + y_i^2) = \sum_j y_j^2(\sum_i A_{ij}) - 2\sum_{i,j} A_{ij} y_i y_j + \sum_i y_i^2(\sum_j A_{ij}) = \sum_j d_j y_j^2 - 2y^T A y + \sum_i d_i y_i^2 = 2y^T D_A y - 2y^T A y = 2y^T(D_A - A)y = 2y^T L y.$)

For the second term (the cross term):

$$-2\sum_{i,j} A_{ij}(y_j - y_i)\widehat{W}_{ij} = -2\sum_{i,j} A_{ij} y_j \widehat{W}_{ij} + 2\sum_{i,j} A_{ij} y_i \widehat{W}_{ij}.$$

The second part of this expression is $2\sum_i y_i \left(\sum_j A_{ij}\widehat{W}_{ij}\right) = 2\sum_i y_i\widehat{\eta}_i = 2y^T\widehat{\eta}$. For the first part, $-2\sum_{i,j} A_{ij}y_j\widehat{W}_{ij}$, we swap indices $i \leftrightarrow j$: $-2\sum_{j,i} A_{ji}y_i\widehat{W}_{ji}$. Since $A_{ji} = A_{ij}$ (undirected graph) and $\widehat{W}_{ji} = -\widehat{W}_{ij}$ (as $\widehat{W}_{ij} = \frac{1}{2}(\widehat{s}_i\widehat{v}_i + \widehat{s}_j\widehat{v}_j)^{\mathsf{T}}(x_j - x_i)$), this becomes: $-2\sum_{i,j} A_{ij}y_i(-\widehat{W}_{ij}) = 2\sum_{i,j} A_{ij}y_i\widehat{W}_{ij} = 2\sum_i y_i \left(\sum_j A_{ij}\widehat{W}_{ij}\right) = 2y^T\widehat{\eta}$. Thus, the entire cross term is $2y^T\widehat{\eta} + 2y^T\widehat{\eta} = 4y^T\widehat{\eta}$.

The third term, $\sum_{i,j} A_{ij}\widehat{W}_{ij}^2$, is a constant with respect to $y$. Therefore, $J(y) = 2y^T Ly - 4y^T\widehat{\eta} +$ constant. Minimizing $J(y)$ subject to $\langle\mathbf{1}, y\rangle = 0$ is equivalent to minimizing $2y^T Ly - 4y^T\widehat{\eta}$ subject to the same constraint, which is further equivalent to minimizing $y^T Ly - 2y^T\widehat{\eta}$ subject to $\langle\mathbf{1}, y\rangle = 0$. This establishes the equivalence stated in (12).

Now, we find the optimal solution. We want to minimize $f(y) = y^T Ly - 2y^T\widehat{\eta}$ subject to $\mathbf{1}^T y = 0$. We form the Lagrangian: $\mathcal{L}(y, \lambda) = y^T Ly - 2y^T\widehat{\eta} - \lambda(\mathbf{1}^T y)$. Taking the derivative with respect to $y$ and setting to zero: $\nabla_y\mathcal{L} = 2Ly - 2\widehat{\eta} - \lambda\mathbf{1} = 0 \implies Ly = \widehat{\eta} + \frac{\lambda}{2}\mathbf{1}$. Let $\lambda' = \lambda/2$. So, $Ly = \widehat{\eta} + \lambda'\mathbf{1}$. For this system to have a solution, the right-hand side must be orthogonal to the null space of $L$. Assuming the graph is connected, the null space of $L$ is spanned by $\mathbf{1}$. Thus, $\mathbf{1}^T(\widehat{\eta} + \lambda'\mathbf{1}) = 0 \implies \mathbf{1}^T\widehat{\eta} + \lambda'\mathbf{1}^T\mathbf{1} = 0 \implies \mathbf{1}^T\widehat{\eta} + n\lambda' = 0$.

We show that $\mathbf{1}^T\widehat{\eta} = 0$: $\mathbf{1}^T\widehat{\eta} = \sum_i \widehat{\eta}_i = \sum_i\sum_j A_{ij}\widehat{W}_{ij}$. Let $M_{ij} = A_{ij}\widehat{W}_{ij}$. Then $M_{ji} = A_{ji}\widehat{W}_{ji} = A_{ij}(-\widehat{W}_{ij}) = -M_{ij}$. The matrix $M$ is anti-symmetric. The sum of all entries of an anti-symmetric matrix is zero, i.e., $\sum_i\sum_j M_{ij} = 0$. Thus, $\mathbf{1}^T\widehat{\eta} = 0$.

Since $\mathbf{1}^T\widehat{\eta} = 0$, the condition $n\lambda' = 0$ implies $\lambda' = 0$ (assuming $n > 0$). The system simplifies to $Ly = \widehat{\eta}$. Since $\mathbf{1}^T\widehat{\eta} = 0$, $\widehat{\eta}$ is in the column space of $L$. The general solution to $Ly = \widehat{\eta}$ is $y = L^+\widehat{\eta} + c\mathbf{1}$ for some constant $c \in \mathbb{R}$, where $L^+$ is the Moore-Penrose pseudo-inverse of $L$. Applying the constraint $\mathbf{1}^T y = 0$: $\mathbf{1}^T(L^+\widehat{\eta} + c\mathbf{1}) = 0 \implies \mathbf{1}^T L^+\widehat{\eta} + c\mathbf{1}^T\mathbf{1} = 0$. Since $L^+\mathbf{1} = \mathbf{0}$ (a property of the pseudo-inverse of a graph Laplacian), we have $\mathbf{1}^T L^+ = (L^+\mathbf{1})^T = \mathbf{0}^T$. So, $\mathbf{0}^T\widehat{\eta} + cn = 0 \implies cn = 0 \implies c = 0$. Therefore, the unique optimal solution satisfying the constraint is $\widehat{y} = L^+\widehat{\eta}$.

### C.2 PROOF OF PROPOSITION 1

For any $t > 0$, let $I_{ij}(t) = 1\{|\theta_i - \theta_j| \le 2t\}$ For $i \ne j$, by independence of $\theta_i$ and $\theta_j$,

$$\mathbb{E}\big[I_{ij}(t)\big] = \iint_{|x-y|\le 2t} f(x)f(y)\,dx\,dy = \int f(x)\left[\int_{x-2t}^{x+2t} f(y)\,dy\right]dx \le 4t\,f_{\max}.$$

Let $U(t) = \frac{1}{\binom{n}{2}}\sum_{i<j} I_{ij}(t)$ which is an order statistic of order 2, with $\mathbb{E}[U(t)] \le 4t\,f_{\max}$. By McDiarmid's (a.k.a. bounded difference) inequality ,

$$\mathbb{P}\big[U(t) \ge 4t f_{\max} + u\big] \le \exp\big(-c\,nu^2\big).$$

Consider the set $\Lambda(t) = \{i < j : |\theta_i - \theta_j| \le 2t\}$. Then, $|\Lambda(t)| = \binom{n}{2}U(t)$. taking $u = \sqrt{\frac{\log(1/\delta)}{cn}}$, we have with probability at least $1 - \delta$,

$$|\Lambda(t)| \le \binom{n}{2}\left(4t f_{\max} + \sqrt{\frac{\log(1/\delta)}{cn}}\right) \le 2t f_{\max}n^2 + C_1\sqrt{\log(1/\delta)}\,n^{3/2}$$

for some constant $C_1$.

Now, let $B(t) = \{i : |y_i - \theta_i| \ge t\}$. Then, $|B(t)| \le n\varepsilon_n^2/t^2$ by Markov inequality (applied to the discrete measure $(1/n, \ldots, 1/n)$).

Let $D$ be the set of discordant pairs, that is, $D = \{i < j : (y_i - y_j)(\theta_i - \theta_j) < 0\}$. A pair $(i, j)$ is discordant only if

- Set $D_1$: either endpoint is in $B(t)$ or
- Set $D_2$: both endpoints are in $[n] \setminus B(t)$ and $|\theta_i - \theta_j| \le 2t$.

that is, $D \subset D_1 \cup D_2$. To see this let $d_{ij}^\theta = \theta_i - \theta_j$ and $d_{ij}^y = y_i - y_j$ and $e_i = y_i - \theta_i$. Then, $d_{ij}^y = d_{ij}^\theta + e_i - e_j$. Suppose that $(i,j)$ is in neither $D_1$ nor $D_2$. Then, $|e_i| < t$ and $|e_j| < t$ and $|d_{ij}^\theta| > 2t$. Since then $|d_{ij}^y - d_{ij}^\theta| < 2t$, $d_{ij}^y$ has the same sign as $d_{ij}^\theta$. Thus, $D_1^c \cap D_2^c \subset D^c$ which is equivalent to the claim.

We have $D_1 \subset \{i < j : i \in B(t) \text{ or } j \in B(t)\}$, hence $|D_1| \le nB(t)$. Also, $D_2 \subset \Lambda(t)$, hence

$$|D| \le n|B(t)| + |\Lambda(t)|$$

and for $n \ge 2$,

$$1 - \tau(y, \theta) = \frac{2|D|}{n(n-1)} \le \frac{4|D|}{n^2} \le \frac{4\varepsilon_n^2}{t^2} + 8tf_{\max} + 4C_1\sqrt{\log(1/\delta)}n^{-1/2}$$

which is the claimed bound. We choose $t$ to make the first to term equal, that is, set $t = (\frac{\varepsilon_n^2}{2f_{\max}})^{1/3}$, and noting that $16/2^{1/3} \le 13$ finishes the proof. The order of the bound is $(\varepsilon_n f_{\max})^{2/3}$.

### C.3 PROOFS OF LEMMA 8 AND LEMMA 6

*Proof.* Fix $i, j$ with $i \sim j$. Let us expand $\gamma(\theta_j^*)$ and $\gamma'(\theta_j^*)$ around $\theta_i^*$. To simplify, write $\delta = \theta_j^* - \theta_i^*$.

$$\gamma(\theta_j^*) = \gamma(\theta_i^*) + \gamma'(\theta_i^*)\delta + \frac{1}{2}g\delta^2$$

$$\gamma'(\theta_j^*) = \gamma'(\theta_i^*) + \widetilde{g}\delta \tag{21}$$

where $g$ and $\widetilde{g}$ are $\gamma''(\cdot)$ evaluated at points between $\theta_i^*$ and $\theta_j^*$. We note that $\|g\|$ and $\|\widetilde{g}\|$ are both $\le \kappa$. Adding $v_i^* = \gamma'(\theta_i^*)$ to the second equation and dividing by 2 we have

$$x_j^* - x_i^* = v_i^*\delta + \frac{1}{2}g\delta^2$$

$$\frac{1}{2}(v_j^* + v_i^*) = v_i^* + \frac{1}{2}\widetilde{g}\delta.$$

Multiplying we obtain,

$$W_{ij}^* = \delta + \frac{1}{2}\langle v_i^*, \widetilde{g}\rangle\delta^2 + \frac{1}{2}\langle v_i^*, g\rangle\delta^2 + \frac{1}{4}\langle g, \widetilde{g}\rangle\delta^3.$$

Note that $|\langle v_i^*, \widetilde{g}\rangle| \le \kappa$ and $|\langle v_i^*, g\rangle| \le \kappa$ and $|\langle g, \widetilde{g}\rangle| \le \kappa^2$. Since $|\delta| \le h^*$, we have

$$|W_{ij}^* - (\theta_j^* - \theta_i^*)| \le \kappa(h^*)^2 + \frac{1}{4}\kappa^2(h^*)^3$$

which proves Lemma 8.

For Lemma 6, from (21), we have $\langle v_j^*, v_i^*\rangle = 1 + \langle \widetilde{g}, v_i^*\rangle\delta$. But then, using $|\langle \widetilde{g}, v_i^*\rangle| \le \kappa$ and $|\delta| \le h^* \le 2\rho + r$ (see the argument after the statement of Lemma 8), we obtain

$$\langle v_j^*, v_i^*\rangle \ge 1 - \kappa(2\rho + r) \ge 1/2$$

using the assumptions $\kappa(2\rho + r) \ll 1$. The proof is complete. $\qquad\square$

### C.4 PROOF OF LEMMA 9

For the first claim, defining $z = y - \theta + \bar{\theta}\mathbf{1}$, we see that $y$ solves (11) iff $z$ solves the same problem with $\widehat{W}$ replaced with $\Delta$. The result now follows from Lemma 1, since $z = L^+\eta$ with $\eta$ is given in the statement.

For the second claim, let $\eta_i = \sum_j A_{ij}\Delta_{ij}$. Using $\widehat{y} = \theta - \bar{\theta} + L^+\eta$, we have

$$\|\widehat{y} - (\theta - \bar{\theta})\| \le \|L^+\|\|\eta\| \le \frac{\|\eta\|}{\lambda_2(L)}$$

Next, $|\eta_i| \le d_i\|\Delta\|_\infty$. It follows that $\frac{1}{n}\|\eta\|^2 \le d_{\text{avg},2}^2\|\Delta\|_\infty^2$. Thus,

$$\frac{1}{n}\|\widehat{y} - (\theta - \bar{\theta})\|^2 \le \left(\frac{d_{\text{avg},2}}{\lambda_2(L)}\right)^2\|\Delta\|_\infty^2$$

Combined with Proposition 1, the result follows.

# D    PSEUDO CODE FOR STAGE

---

**Algorithm 1:** STAGE: Spectral Tangent Alignment and Geometric Embedding for Linear Seriation

---

**Input:** Data points $\mathscr{X}_n = \{x_1, \ldots, x_n\} \subset \mathbb{R}^d$; neighborhood radius $r > 0$; minimum neighbors $d_{\min} \geq 2$.

**Output:** Permutation $\phi$ of $\{1, \ldots, n\}$ representing the linear order.

```
/* Stage 1:  Local Tangent Estimation                          */
```
**foreach** $x_i \in \mathscr{X}_n$ **do**

    Define neighborhood $\mathcal{N}(x_i, r) = \{x_j \in \mathscr{X}_n \mid \|x_j - x_i\|_2 \leq r\}$.

    Let $d_i = |\mathcal{N}(x_i, r)|$.

    **if** $d_i \geq d_{\min}$ **then**

        Compute local mean: $\widehat{\mu}_i = \frac{1}{d_i} \sum_{x_j \in \mathcal{N}(x_i, r)} x_j$.

        Compute local covariance: $\widehat{\Sigma}_i = \frac{1}{d_i} \sum_{x_j \in \mathcal{N}(x_i, r)} (x_j - \widehat{\mu}_i)(x_j - \widehat{\mu}_i)^{\intercal}$.

        $\widehat{v}_i \leftarrow$ normalized principal eigenvector of $\widehat{\Sigma}_i$.

    **else**

        $\widehat{v}_i \leftarrow \mathbf{0}$ (or handle as missing/unreliable).        *// Tangent undefined*

```
/* Stage 2:  Orientation Alignment                             */
```
Construct affinity matrix $\widehat{Q} \in \mathbb{R}^{n \times n}$ where:

$$\widehat{Q}_{ij} = \begin{cases} \langle \widehat{v}_i, \widehat{v}_j \rangle & \text{if } i \sim j \text{ and } \widehat{v}_i, \widehat{v}_j \neq \mathbf{0} \\ 0 & \text{otherwise} \end{cases}$$
        *// $i \sim j \iff \|x_i - x_j\|_2 \leq r$*

Compute $\widehat{u} \leftarrow$ principal eigenvector of $\widehat{Q}$.

**foreach** $i \in \{1, \ldots, n\}$ **do**

    $\widehat{s}_i \leftarrow \text{sign}(\widehat{u}_i)$ (or 1 if $\widehat{u}_i = 0$ or $\widehat{v}_i = \mathbf{0}$).

Let $\widetilde{v}_i = \widehat{s}_i \widehat{v}_i$ be the oriented tangents.

```
/* Stage 3:  Global Embedding via Least Squares                */
```
**foreach** *pair* $(i, j)$ *such that* $i \sim j$ **do**

    $\widehat{W}_{ij} \leftarrow \frac{1}{2}(\widetilde{v}_i + \widetilde{v}_j)^{\intercal}(x_j - x_i)$.

Solve for the embedding $\widehat{y} \in \mathbb{R}^n$:

$$\widehat{y} = \underset{y \in \mathbb{R}^n \colon \langle \mathbf{1}, y \rangle = 0}{\text{argmin}} \sum_{i,j \colon i \sim j} \left( y_j - y_i - \widehat{W}_{ij} \right)^2.$$

```
/* Stage 4:  Final Ordering                                    */
```
$\phi \leftarrow \text{argsort}_{j \in \{1, \ldots, n\}} \widehat{y}_j$.

**return** $\phi$

---

# E    ADDITIONAL EXPERIMENTS

## E.1    2-D EXPERIMENTS

Our 2-D dataset consists of points sampled from a rotated sine curve embedded in $\mathbb{R}^2$. We introduce additive Gaussian noise with standard deviation $\sigma$ varying from 0.1 to 0.5 (in increments of 0.1). Experiments were run for sample sizes $n \in \{500, 1000, 3000\}$. For each configuration (noise level and sample size), we conducted 10 independent trials, reporting the average Kendall's $\tau$ and its standard deviation in Table 3.

The results in Table 3 demonstrate that STAGE consistently achieves the highest absolute Kendall's $\tau$ values across all tested noise levels and sample sizes, indicating superior order recovery compared to the baseline methods. For clarity, Kendall's $\tau$ values in Table 3 are multiplied by 100.

Table 3: Comparison of Methods on 2-D sine curve, Kendall's $\tau$ and std. err. values multiplied by 100

| Sample | Method | $\sigma = 0.1$ | $\sigma = 0.2$ | $\sigma = 0.3$ | $\sigma = 0.4$ | $\sigma = 0.5$ |
|---|---|---|---|---|---|---|
| 500 | Fiedler Vector | 61.39 (5.06) | 73.0 (5.51) | 74.1 (5.89) | 74.0 (6.2) | 76.55 (3.83) |
| | 1D UMAP | 70.59 (13.25) | 69.35 (13.08) | 71.21 (16.11) | 66.32 (15.94) | 73.52 (7.34) |
| | 1D t-SNE | 97.82 (0.11) | 95.38 (0.21) | 92.76 (0.21) | 90.17 (0.41) | 87.2 (0.75) |
| | Recanati | 97.65 (0.1) | 95.35 (0.18) | 92.46 (0.43) | 87.92 (1.61) | 81.54 (5.34) |
| | STAGE | **97.85 (0.04)** | **95.41 (0.20)** | **93.03 (0.20)** | **90.26 (0.38)** | **87.34 (0.39)** |
| 1000 | Fiedler Vector | 65.63 (2.85) | 74.98 (3.94) | 78.31 (3.79) | 76.8 (1.81) | 68.4 (22.68) |
| | 1D UMAP | 73.68 (12.83) | 77.33 (11.5) | 73.19 (14.24) | 67.67 (14.99) | 74.71 (12.53) |
| | 1D t-SNE | 97.69 (0.09) | 95.2 (0.15) | 92.56 (0.11) | 90.0 (0.37) | 87.08 (0.49) |
| | Recanati | 97.64 (0.08) | 95.23 (0.11) | 92.47 (0.35) | 84.53 (5.0) | 78.57 (3.2) |
| | STAGE | **97.82 (0.06)** | **95.47 (0.09)** | **93.03 (0.14)** | **90.34 (0.32)** | **87.51 (0.28)** |
| 3000 | Fiedler Vector | 66.49 (3.67) | 77.49 (2.54) | 78.06 (3.03) | 78.8 (2.31) | 78.16 (1.44) |
| | 1D UMAP | 80.88 (7.65) | 79.99 (8.43) | 86.69 (4.82) | 81.6 (5.76) | 75.27 (5.54) |
| | 1D t-SNE | 97.3 (0.38) | 95.08 (0.12) | 92.56 (0.11) | 89.88 (0.22) | 87.21 (0.22) |
| | Recanati | 97.6 (0.02) | 95.22 (0.09) | 92.32 (0.69) | 85.11 (2.96) | 75.04 (3.04) |
| | STAGE | **97.76 (0.04)** | **95.44 (0.07)** | **92.92 (0.11)** | **90.30 (0.18)** | **87.54 (0.22)** |

## E.2 HIGHER DIMENSION EXPERIMENTS

Here, we provide additional experiments to demonstrate the performance of the STAGE algorithm. In Figure 5 we display an example random Fourier curve in 3 dimensions. Table 4 shows the full results of our high-dimensional experiments, across a range of sample sizes. Figure 6 shows the pairplots as shown in Figure 2 in the main text, along with the true curves to demonstrate the effectiveness of the denoising. In Figures 7 and 8 we demonstrate the increase in performance and runtime of STAGE and t-SNE as the neighborhood size and perplexity increase, respectively. These plots clearly show both the superior accuracy of STAGE and the lower computational cost relative to t-SNE. We perform similar comparisons to additional embedding methods ISOMAP and LLE in Figures 9, 10 and 11. We compare the methods with $n = 500, 1000, 2000$ and $d = 50, 100, 200$, with 10 replicates per setting and $\sigma = 2$. These experiments show that we do somewhat better at recovery than ISOMAP, particularly at higher noise levels and dimensionality and greatly outperform LLE. LLE is faster (but lacks accuracy), whereas STAGE is slightly slower than ISOMAP at lower sample sizes, but faster at $n = 2000$. LLE also seems much more sensitive to the choice of neighborhood size. We further explore the differences in compute time with respect to all the methods in Table 5, where we see that our method is comparable to UMAP, roughly 2× faster than arguably the strongest competitor, t-SNE ($\sim$2.3× at $n = 1000$, $\sim$2.2× at $n = 2000$), and $\sim$3× faster than Recanati and Fiedler at $n = 1000$, rising to $\sim$4–5× at $n = 2000$. Finally, in order to better understand the effect of curvature and noise on our model, we apply our method to unit-length arcs drawn from circles whose curvatures range from 0 to 5, and with $\sigma$ varying between 0 and 5. We ran these simulations for $n = 1000$ and $n = 2000$, where for each combination of $\sigma$ and $\kappa$ we embedded the circle in $d = 10, 50, 500$ and averaged the results of 20 replicates. In Figure 12, we provide a heatmap of the average $\tau$ over $\kappa$ and $\sigma$ where we see that as $\kappa$ increases or as $d$ increases, so does the sensitivity to noise.

## E.3 SCRNA-SEQ DATA

Here we include some further analysis of the scRNA-Seq datasets discussed in the main paper. In particular, we show the distribution of the residuals and a degree analysis of the resulting neighborhood graphs. In Figures 13, 14 and 15 we see that for the most part, the residuals look normal. There is some deviation in the Packer seam cell data, which has some heavier tails, but nonetheless our performance for this dataset is quite strong. Finally, we tested for heavy tails by fitting a power-law tail and comparing it to thin-tailed alternatives using the Clauset–Shalizi–Newman procedure Clauset et al. (2009) (Python powerlaw, discrete degrees). The method (i) selects $x_{min}$ by minimizing the KS distance, (ii) fits the power-law exponent on $d \geq x_{min}$, and (iii) uses log-likelihood ratio (LLR) tests to compare power-law vs log-normal and exponential tails. The results can be seen in Table 6. Negative LLR means the alternative fits better; the reported $p$ is the p-value for that comparison. We also summarize overall concentration via CV (SD/mean) and the 99th-percentile/median ratio.

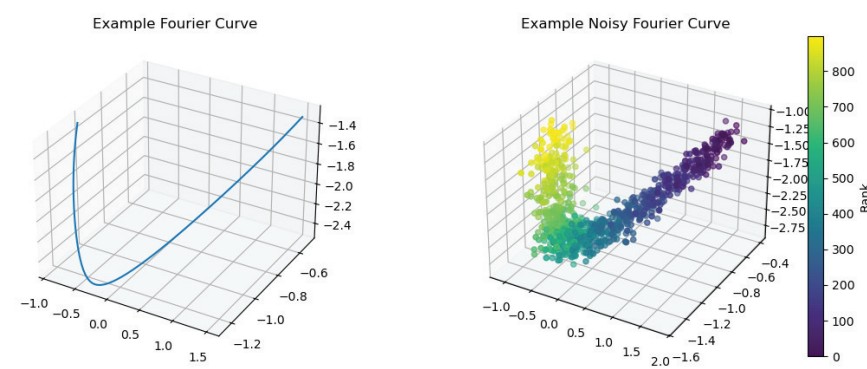

Figure 5: (Left) First 3 dimensions of true Fourier curve, (Right) First 3 dimensions of perturbed Fourier curve, colored by order

We also include an analysis of results based on different factors of the smallest connectivity radius for each dataset (denoted $r_0$) in Figure 16. These demonstrate that for some factor of the minimal connectivity radius, we get good results with STAGE, and past that point the performance drops off, but not significantly. The behavior is not as well demonstrated in the last dataset - a direction for further investigation might be to understand this behavior.

# F LARGE LANGUAGE MODEL (LLM) USAGE

We used LLMs (GPT-5 and Gemini) to help with writing code and polishing the writing of the paper.

Table 4: Comparison of Methods on high-dimensional Fourier curves, Kendall's $\tau$ and std. err. values multiplied by 100

| Dim. | Sample | Method | $\sigma = 0.5$ | $\sigma = 1$ | $\sigma = 1.5$ | $\sigma = 2$ |
|------|--------|--------|---------------|--------------|---------------|--------------|
| 50 | 500 | Fiedler Vector | 81.7 (8.62) | 77.55 (3.51) | 60.03 (7.39) | 28.82 (17.16) |
| | | 1D UMAP | 88.27 (0.97) | 75.08 (2.62) | 60.42 (4.55) | 38.93 (14.45) |
| | | 1D t-SNE | 90.62 (0.87) | 75.53 (15.43) | 68.52 (3.66) | 57.77 (5.52) |
| | | Recanati | 89.86 (1.14) | 79.31 (1.67) | 70.11 (1.85) | 62.95 (2.21) |
| | | STAGE | **91.11 (0.9)** | **82.06 (1.59)** | **72.28 (2.19)** | **63.79 (2.73)** |
| | 1000 | Fiedler Vector | 84.58 (6.86) | 77.45 (3.21) | 44.81 (18.28) | 26.89 (8.16) |
| | | 1D UMAP | 88.43 (1.0) | 76.02 (2.0) | 63.1 (3.81) | 47.59 (6.24) |
| | | 1D t-SNE | 90.42 (0.79) | 80.6 (1.58) | 71.36 (1.85) | 60.71 (4.97) |
| | | Recanati | 89.87 (1.2) | 79.61 (1.38) | 71.96 (1.76) | 63.53 (2.81) |
| | | STAGE | **90.92 (0.78)** | **81.69 (1.44)** | **73.11 (1.91)** | **64.06 (3.4)** |
| | 2000 | Fiedler Vector | 85.47 (4.53) | 76.91 (3.53) | 49.49 (11.23) | 33.22 (14.77) |
| | | 1D UMAP | 88.6 (0.67) | 77.08 (1.89) | 64.5 (3.41) | 51.17 (3.5) |
| | | 1D t-SNE | 90.51 (0.63) | 81.03 (1.43) | 71.53 (2.75) | 61.51 (3.16) |
| | | Recanati | 90.05 (1.03) | 80.1 (1.35) | 71.6 (1.86) | **64.39 (2.35)** |
| | | STAGE | **90.91 (0.66)** | **81.76 (1.47)** | **72.26 (2.36)** | 64.04 (2.58) |
| 100 | 500 | Fiedler Vector | 75.88 (12.19) | 82.34 (1.84) | 69.3 (3.69) | 48.96 (8.28) |
| | | 1D UMAP | 90.66 (0.66) | 80.5 (1.31) | 66.4 (2.19) | 55.4 (4.61) |
| | | 1D t-SNE | 92.89 (0.43) | 69.86 (23.17) | 75.24 (4.65) | 65.74 (6.4) |
| | | Recanati | 92.32 (0.59) | 83.41 (0.65) | 76.55 (0.92) | 70.23 (1.81) |
| | | STAGE | **93.42 (0.39)** | **86.46 (0.72)** | **79.23 (1.18)** | **72.74 (2.27)** |
| | 1000 | Fiedler Vector | 82.12 (9.06) | 82.06 (2.84) | 66.65 (6.98) | 41.75 (18.69) |
| | | 1D UMAP | 90.93 (0.75) | 80.82 (1.5) | 70.61 (1.56) | 55.6 (6.88) |
| | | 1 -D t-SNE | 92.91 (0.49) | 81.58 (10.42) | 78.03 (1.76) | 67.84 (4.79) |
| | | Recanati | 92.55 (0.66) | 84.11 (0.75) | 77.55 (0.94) | 71.61 (1.08) |
| | | STAGE | **93.41 (0.45)** | **86.47 (0.72)** | **79.59 (1.23)** | **72.38 (1.54)** |
| | 2000 | Fiedler Vector | 83.1 (12.9) | 83.34 (1.55) | 63.0 (13.29) | 44.9 (11.3) |
| | | 1D UMAP | 90.89 (0.45) | 81.72 (1.18) | 71.14 (2.07) | 58.85 (4.81) |
| | | 1D t-SNE | 92.93 (0.33) | 85.66 (0.86) | 78.28 (1.49) | 70.05 (3.35) |
| | | Recanati | 92.77 (0.47) | 84.51 (0.76) | 77.9 (1.03) | 72.07 (1.39) |
| | | STAGE | **93.37 (0.29)** | **86.46 (0.85)** | **79.47 (1.39)** | **72.27 (1.89)** |
| 200 | 500 | Fiedler Vector | 40.39 (15.15) | 86.18 (4.95) | 79.75 (3.77) | 65.57 (9.33) |
| | | 1D UMAP | 93.17 (0.31) | 85.89 (0.7) | 77.18 (1.39) | 66.43 (2.57) |
| | | 1D t-SNE | 95.18 (0.19) | 89.73 (0.58) | 77.01 (18.78) | 76.19 (3.27) |
| | | Recanati | 94.89 (0.21) | 87.17 (0.43) | 82.1 (0.92) | 76.87 (0.87) |
| | | STAGE | **95.63 (0.22)** | **90.99 (0.27)** | **86.22 (0.66)** | **80.98 (0.84)** |
| | 1000 | Fiedler Vector | 62.64 (30.52) | 84.71 (9.72) | 81.12 (2.29) | 64.05 (6.77) |
| | | 1D UMAP | 85.22 (23.88) | 85.86 (0.88) | 78.01 (0.75) | 68.86 (1.46) |
| | | 1D t-SNE | 95.13 (0.17) | 89.96 (0.32) | 73.14 (20.3) | 76.26 (4.62) |
| | | Recanati | 95.13 (0.16) | 87.62 (0.42) | 82.92 (0.6) | 78.12 (0.55) |
| | | STAGE | **95.58 (0.14)** | **90.95 (0.36)** | **86.03 (0.55)** | **81.31 (0.52)** |
| | 2000 | Fiedler Vector | 68.8 (22.04) | 88.7 (0.37) | 81.2 (1.1) | 66.52 (4.1) |
| | | 1D UMAP | 93.29 (0.33) | 86.35 (0.46) | 78.89 (0.62) | 70.17 (1.03) |
| | | 1D t-SNE | 95.01 (0.08) | 89.94 (0.26) | 84.98 (0.28) | 77.47 (4.18) |
| | | Recanati | 95.17 (0.12) | 87.84 (0.21) | 83.41 (0.34) | 78.88 (0.49) |
| | | STAGE | **95.47 (0.09)** | **90.77 (0.24)** | **86.12 (0.28)** | **81.18 (0.56)** |

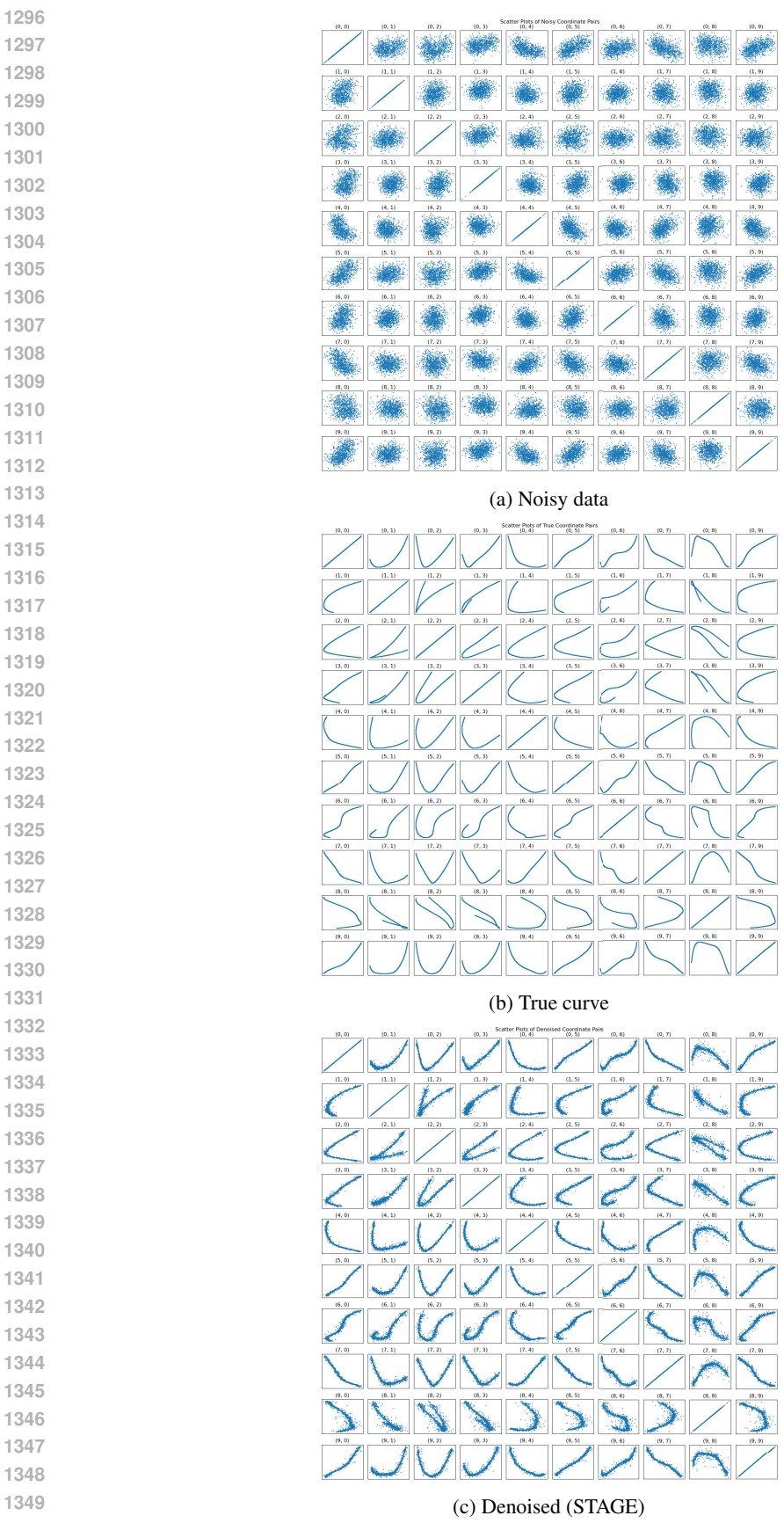

(a) Noisy data

(b) True curve

(c) Denoised (STAGE)

Figure 6: Pairplots of the first 10 dimensions: (a) original noisy observations, (b) underlying curve, and (c) denoised points produced by STAGE.

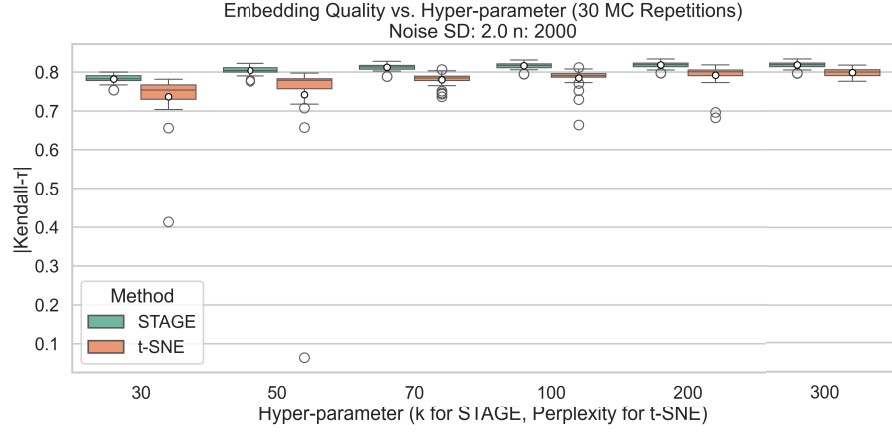

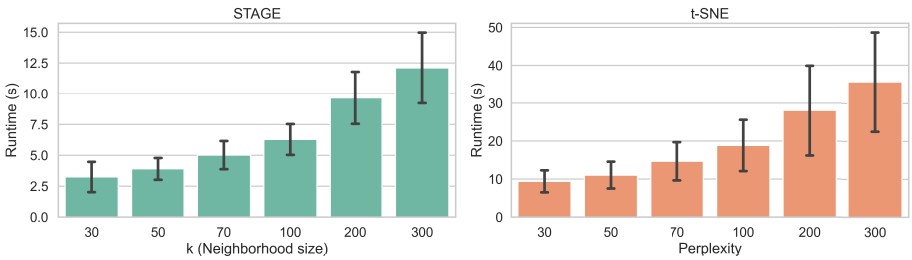

Figure 7: (Top) Order recovery of STAGE vs. t-SNE as neighborhood size/perplexity increases, (Bottom) Runtime of STAGE vs. t-SNE as neighborhood size/perplexity increases ($y$-axis scales differ)

Table 5: Comparison of runtimes (in seconds) on high-dimensional Fourier curves

| Dim. | Sample | Fiedler Vector | 1D UMAP | 1D t-SNE | Recanati | STAGE |
|------|--------|----------------|---------|----------|----------|-------|
| 50   | 500    | 0.4651         | 0.5631  | 0.7526   | 0.5212   | 0.2468 |
|      | 1000   | 1.9515         | 0.5643  | 1.8137   | 1.775    | 0.4894 |
|      | 2000   | 8.7202         | 1.5007  | 2.5898   | 6.8389   | 1.1483 |
| 100  | 500    | 0.497          | 0.2493  | 0.7831   | 0.5239   | 0.2769 |
|      | 1000   | 2.0782         | 0.5483  | 1.5271   | 1.8017   | 0.5879 |
|      | 2000   | 9.5623         | 1.6735  | 3.5941   | 7.2733   | 1.4597 |
| 200  | 500    | 0.4798         | 0.2378  | 1.5625   | 0.5283   | 0.3081 |
|      | 1000   | 2.1054         | 0.6303  | 1.5625   | 1.8157   | 0.6904 |
|      | 2000   | 8.7893         | 1.5472  | 3.605    | 6.8337   | 1.6502 |

Table 6: Tail tests & concentration (per dataset)

| Dataset | LLR (vs LN) | $p$ (vs LN) | LLR (vs exp) | $p$ (vs exp) | CV | 99th/median |
|---------|-------------|-------------|--------------|--------------|-----|-------------|
| Saelens | -17.67 | 6.77e-70 | -29.01 | 5.45e-185 | 0.42 | 1.27 |
| Packer Seam Cells | -15.51 | 2.86e-54 | -16.78 | 3.48e-63 | 0.75 | 3.81 |
| Packer Hypodermis | -37.34 | 3.33e-305 | -79.19 | 0 | 0.50 | 1.38 |

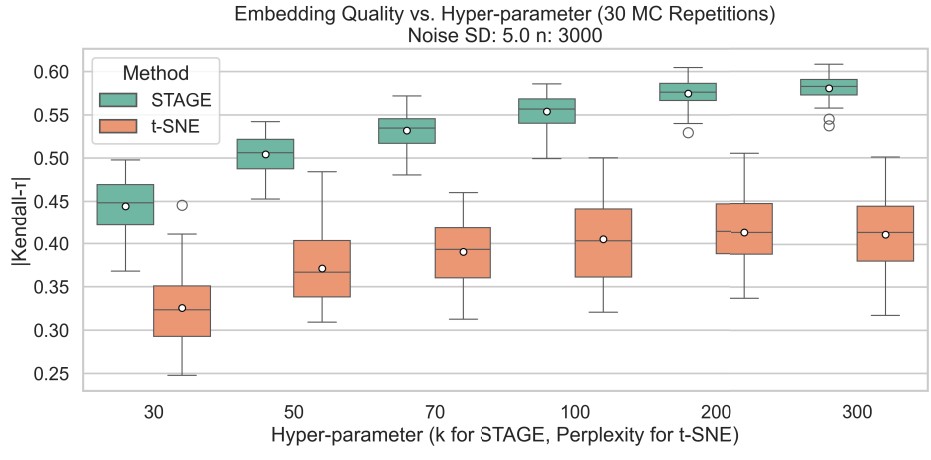

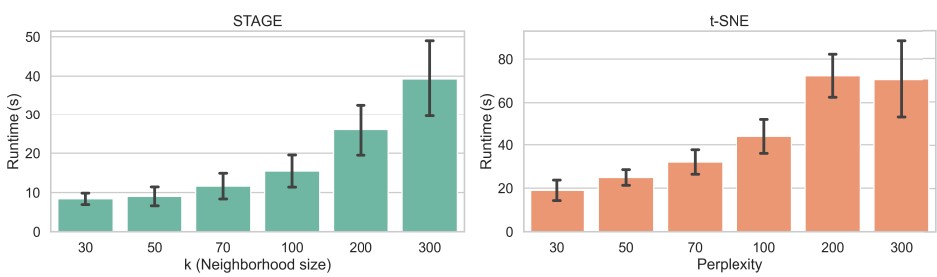

Figure 8: (Top) Order recovery of STAGE vs. t-SNE as neighborhood size/perplexity increases, (Bottom) Runtime of STAGE vs. t-SNE as neighborhood size/perplexity increases ($y$-axis scales differ)

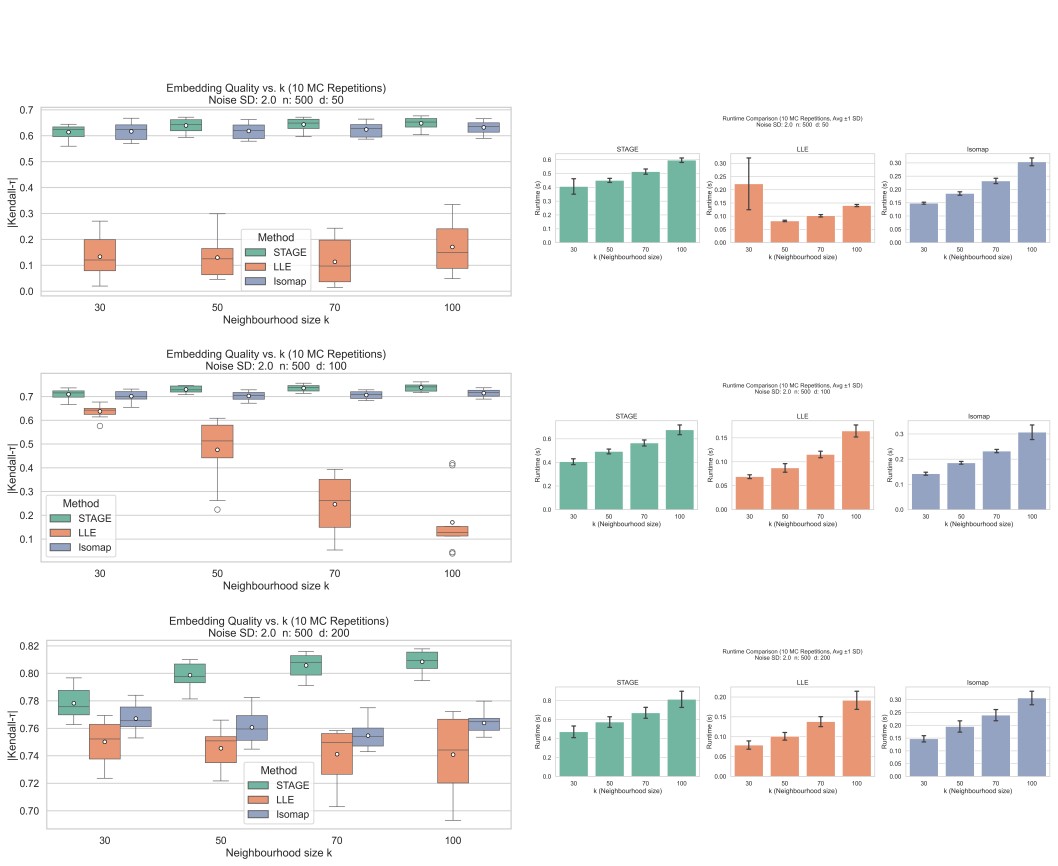

Figure 9: Kendall's $\tau$ and runtime comparison of STAGE vs. ISOMAP vs. LLE with n = 500

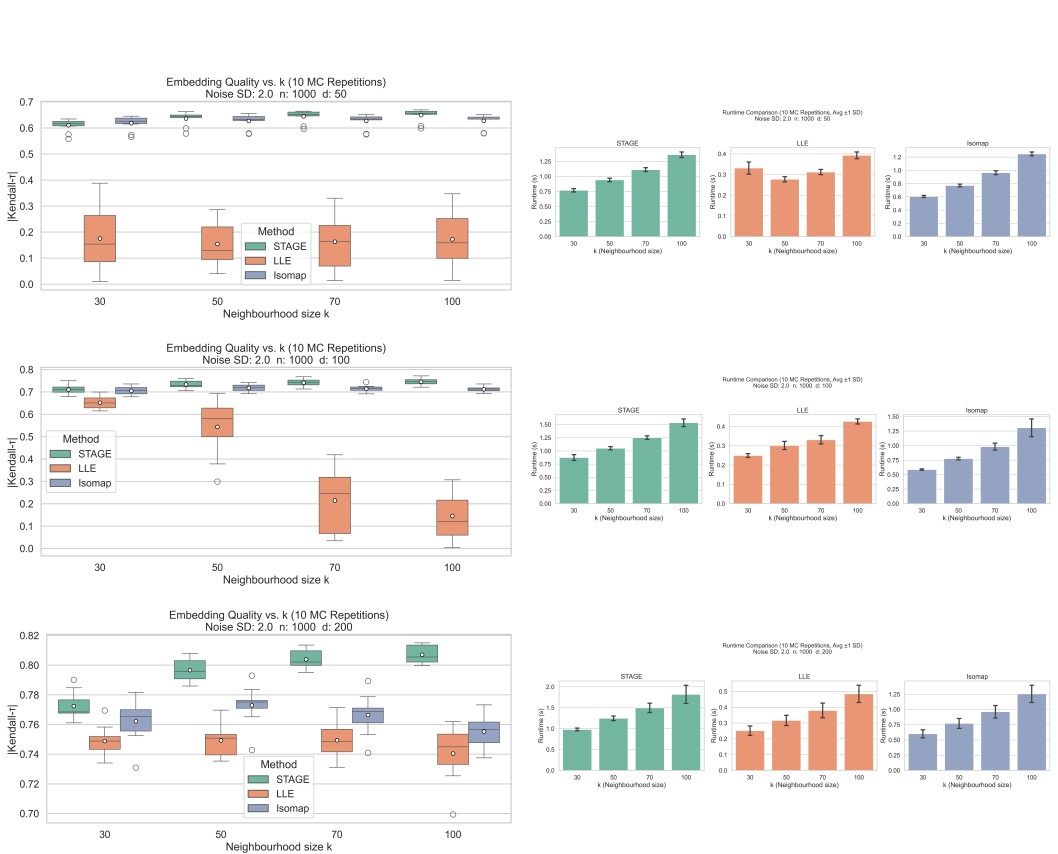

Figure 10: Kendall's $\tau$ and runtime comparison of STAGE vs. ISOMAP vs. LLE with n = 1000

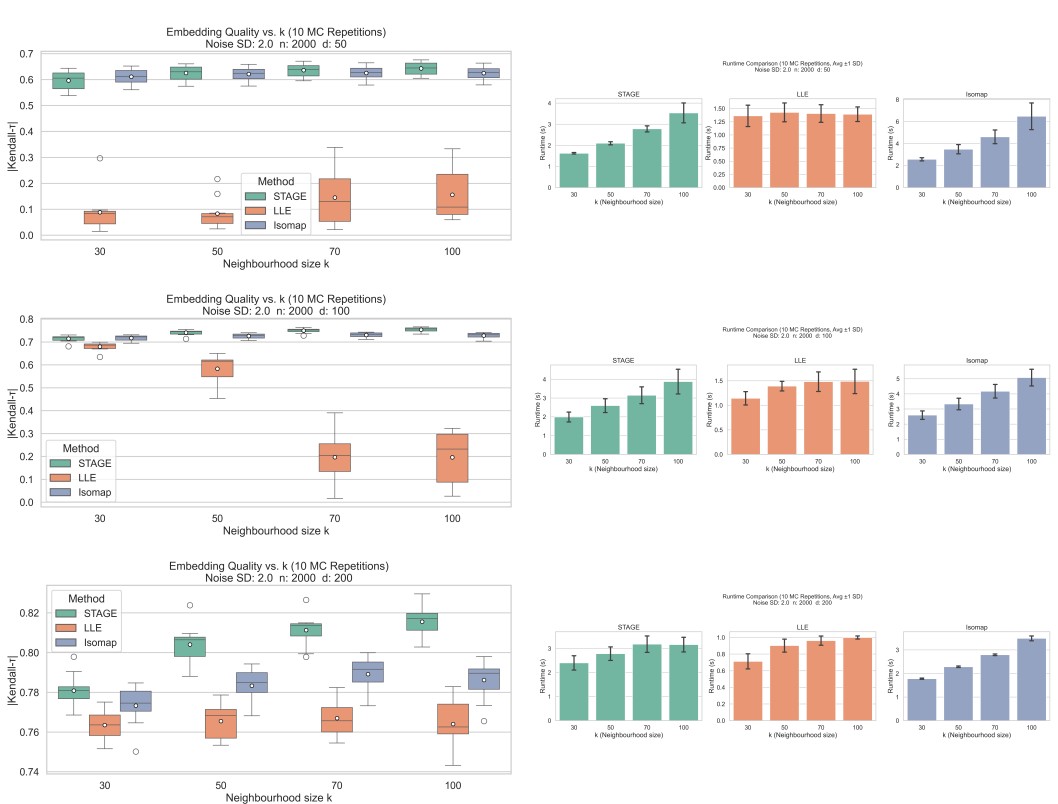

Figure 11: Kendall's $\tau$ and runtime comparison of STAGE vs. ISOMAP vs. LLE with n = 2000

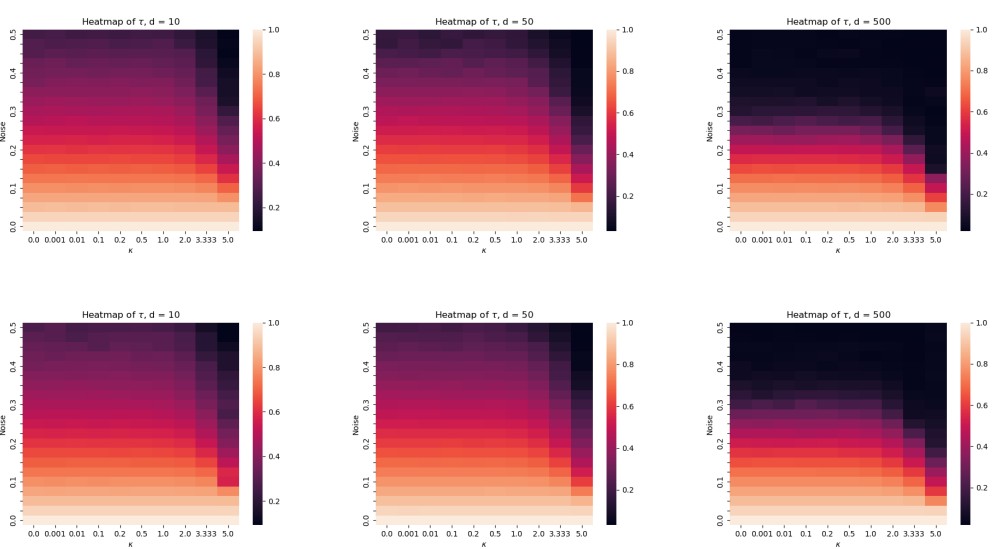

Figure 12: Heatmap of $\tau$ (averaged over 20 runs) for various values of $\kappa$ and $\sigma$. Top row: $n = 1000$ with $d = 10, 50, 500$ from left to right. Bottom row: $n = 2000$ with $d = 10, 50, 500$ from left to right.

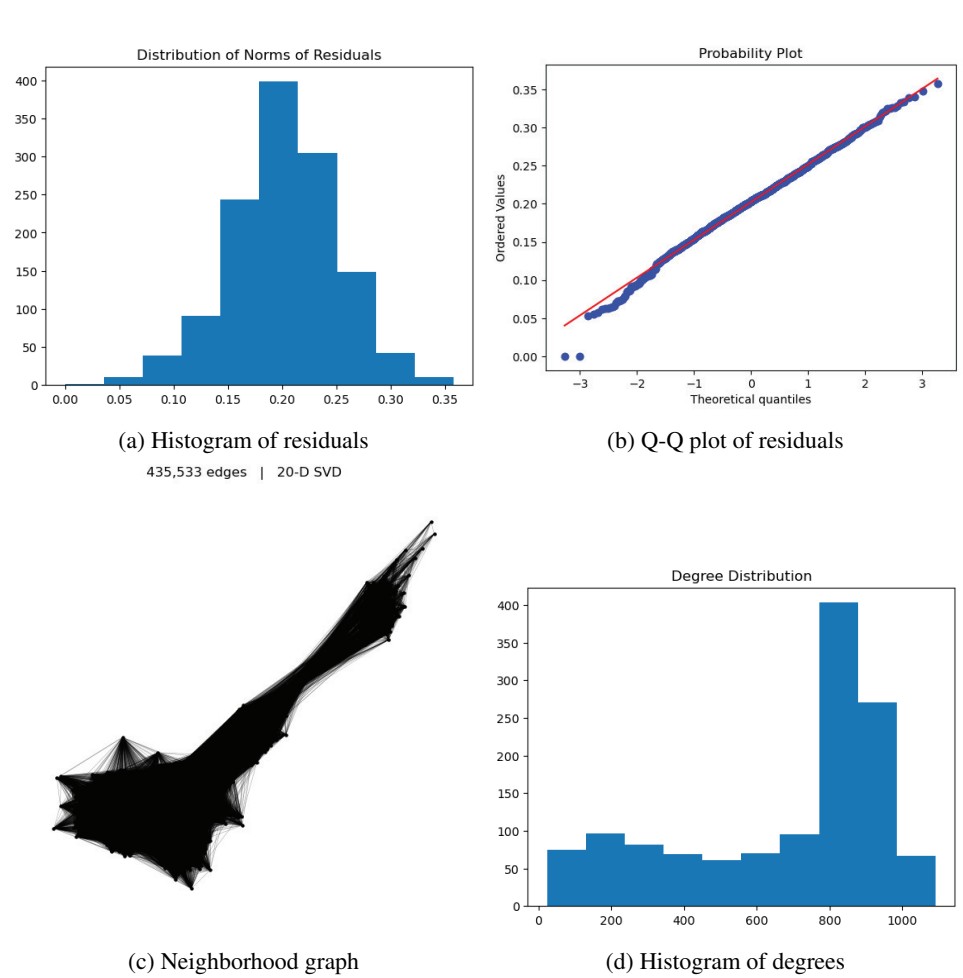

(a) Histogram of residuals        (b) Q-Q plot of residuals

(c) Neighborhood graph        (d) Histogram of degrees

Figure 13: Residual and degree distributions of the STAGE algorithm applied to the Saelens dataset: (a) histogram of residuals, (b) Q-Q plot of residuals, (c) neighborhood graph, and (d) histogram of degrees

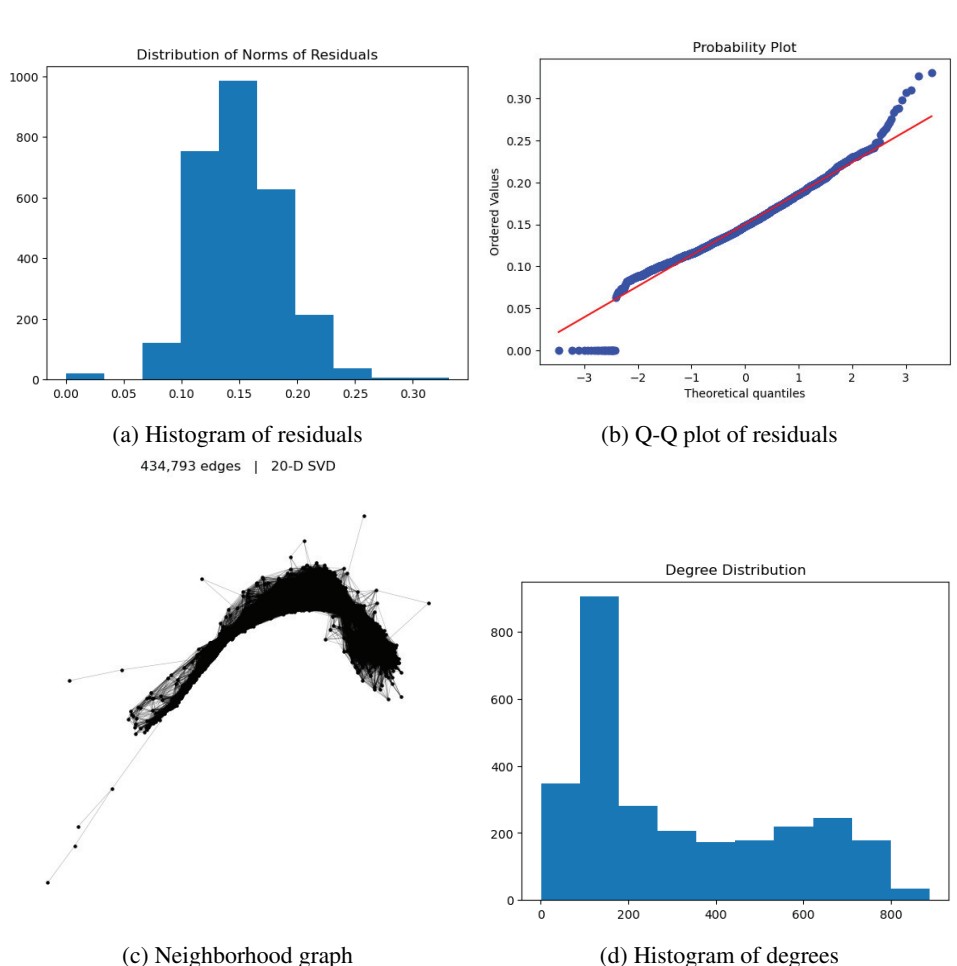

(a) Histogram of residuals

(b) Q-Q plot of residuals

(c) Neighborhood graph

(d) Histogram of degrees

Figure 14: Residual and degree distributions of the STAGE algorithm applied to the Packer seam cell dataset: (a) histogram of residuals, (b) Q-Q plot of residuals, (c) neighborhood graph, and (d) histogram of degrees

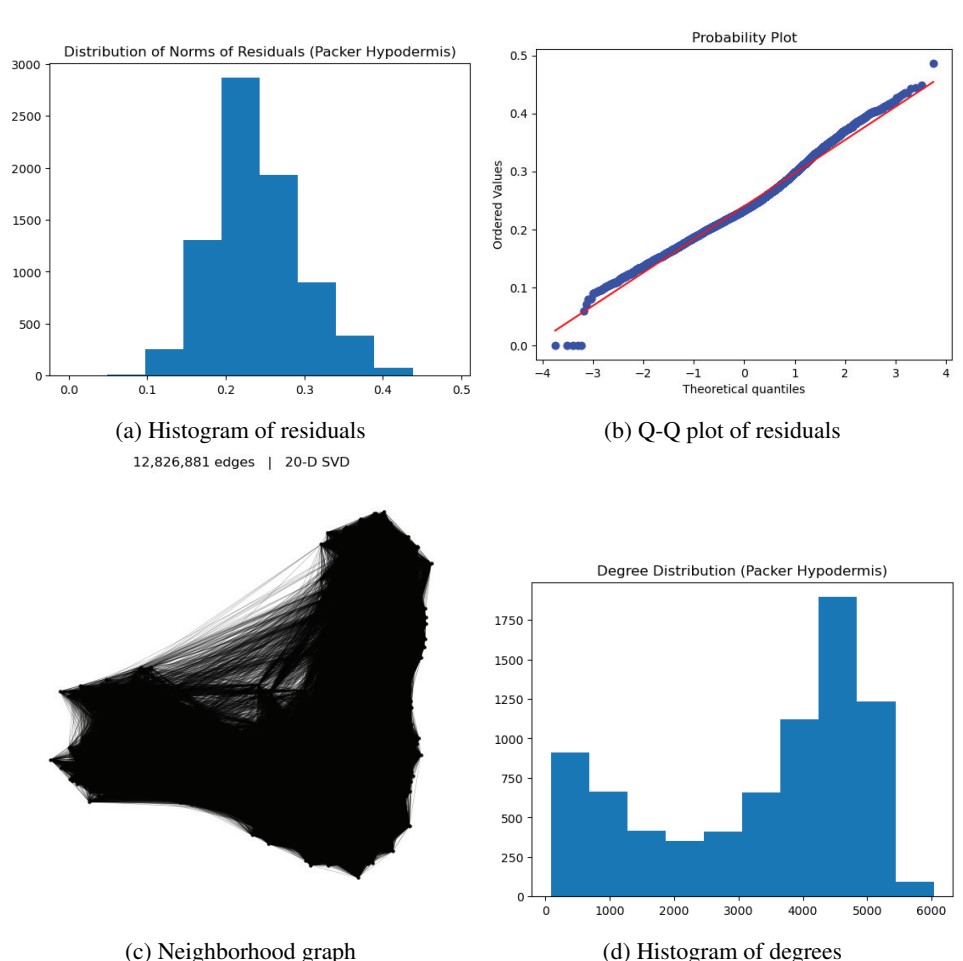

(a) Histogram of residuals

(b) Q-Q plot of residuals

(c) Neighborhood graph

(d) Histogram of degrees

Figure 15: Residual and degree distributions of the STAGE algorithm applied to the Packer hypodermis dataset: (a) histogram of residuals, (b) Q-Q plot of residuals, (c) neighborhood graph, and (d) histogram of degrees

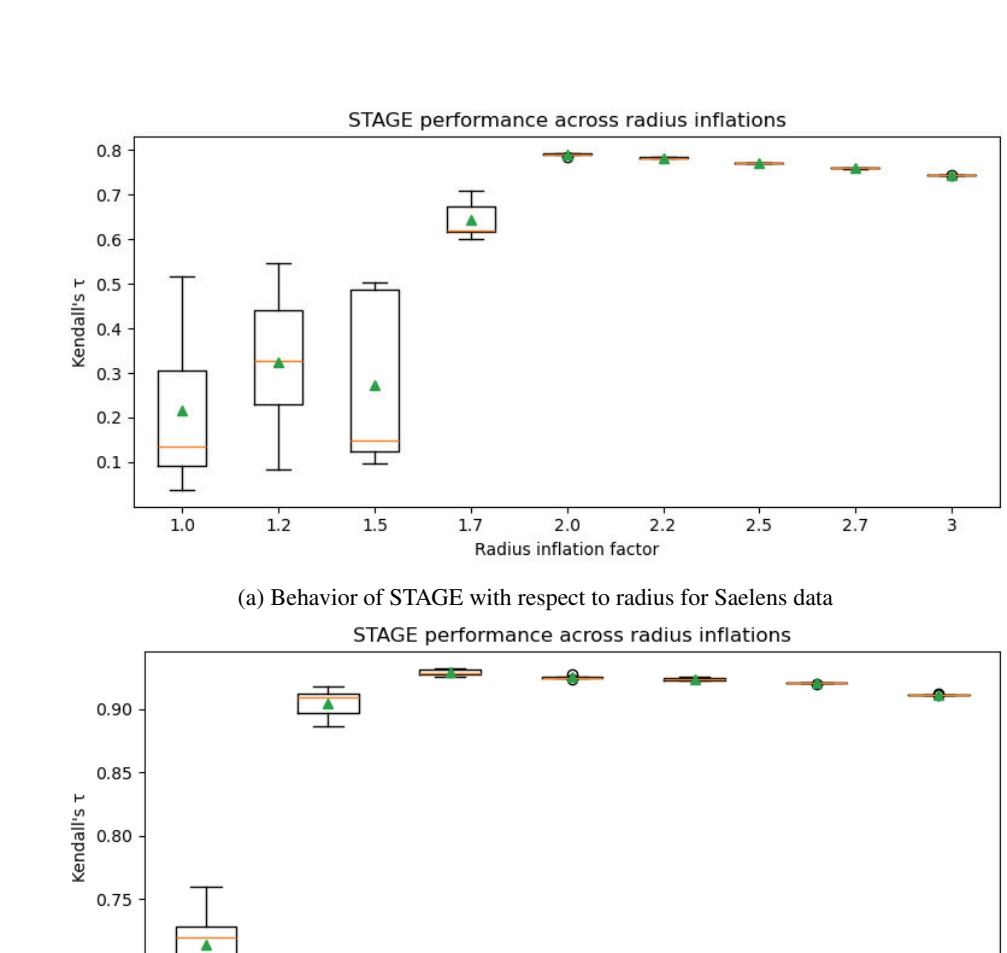

(a) Behavior of STAGE with respect to radius for Saelens data

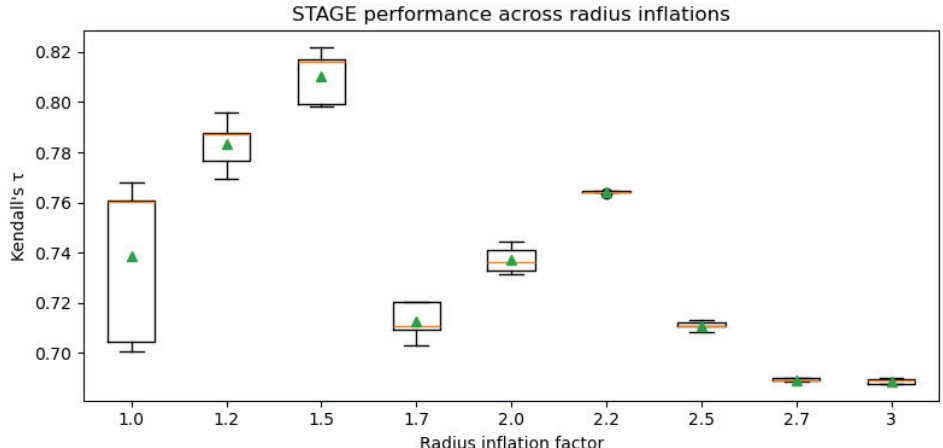

(b) Behavior of STAGE with respect to radius for Packer seam cell data

(c) Behavior of STAGE with respect to radius for Packer hypodermis data

Figure 16: Behavior of STAGE with respect to neighborhood radius

