# OpenReview forum: "Trajectory Seriation via Spectral Tangent Alignment and Global Embedding"
_ICLR.cc/2026/Conference — Submitted to ICLR 2026_

### Official Review · Reviewer_LY8v · 2025-10-21

**Soundness:** 2
**Presentation:** 4
**Contribution:** 3
**Rating:** 4
**Confidence:** 4

**Summary:**

This paper presents a new algorithm for linear seriation of data in R^d. i.e. assigning an order to a point cloud when we assume the points roughly follow a one-dimensional curve.

The algorithm is divided into three steps:
1. Estimate the local tangent (up to sign) at each data point using local PCA.
2. Pick signs for all the tangent vectors so that they are globally compatible.
3. Find the order of the points by solving a least-squares problems that assigns values to the points so that local differences approximate the oriented tangent vectors.

The authors test their method and compare it to the Fiedler vector method of Atkins et al. [1], t-SNE [2], UMAP [3] and Recanati et al. (2017), demonstrating superior performance. The paper contains a proof that shows that under certain assumptions, the resulting ordering is highly correlated with the ground-truth order.

**Strengths:**

* The paper is very clear and a pleasure to read.
* The described method is short and elegant.
* Empirically, the proposed method (slightly) outperforms existing method in terms of error and runtime.
* The paper contains a theorem that shows that, with some assumptions on the data distribution, with probability > 0.99, the order that is found is close to the truth (in a Kendall tau sense). However, I believe there is an issue with the current theorem statement (see below).

**Weaknesses:**

* As written, there appears to be an error in Theorem 1. The curve is required to be parameterized by an injective function of bounded curvature, which prevents self-intersections. However, it can be *close* to self-intersection. This could cause the local tangent estimation to fail because it would mix points from two branches of the curve.  While a major error in the theorem statement, I don't think it is difficult to fix by adding an assumption on the reach of the manifold (a formal notion of the distance to self-intersection). I think it may enough to require that the radius r used in the algorithm be smaller than reach(\gamma)-2\rho, but please check this carefully.

* The paper claims that methods based on a similarity matrix are sensitive to outliers and thus "can lead to a loss of finer geometric information inherent in the point cloud coordinates" but I don't think this claim is sufficiently supported in the manuscript. In particular, there are no theoretical results that support it. The empirical evaluations show that Recanati (2017) actually performs very well in the empirical comparisons (Table 1, Table 2, Table 3, Table 4), even in high dimensions such as 200.

**Questions:**

* Why \alpha=2.3 in the simulation in Section 5.1? This value raises suspicion of cherry-picking.

* More of a comment than a question: in addition to the issue raised w.r.t. Theorem 1, I would suggest that you modify the theorem statement such that it holds w.p. > 1-\epsilon for any epsilon. I believe you could even guarantee convergence to the true order w.h.p. under appropriate conditions if you are willing to consider a curve-orthogonal noise model. i.e. where the noise at each point is a Gaussian in the d-1 subspace orthogonal to the tangent of the curve at the point.

---

> ### Author Response · Authors · 2025-11-28
>
> Thank you for your positive assessment and valuable feedback. Please find our responses to your comments below.
>
> **About self intersection:**
> Thank you for pointing this out. You are absolutely right. We missed an assumption. We added Assumption 4 to prevent near self-intersection. There was a subtle bug in applying Lemma 3 in the proof, now corrected: Assumption 4 allows us to verify the condition $|\theta - \theta_0| \le \pi/\kappa$ needed in Lemma 3.
>
> Whether Assumption 4 follows from a condition related to reach is unclear, but the current form is straightforward, easy to understand, and exactly what we need.
>
>
> **On information loss in similarity-based approaches:** Thanks for the comment. In the revised paper, we have added Appendix A, discussing what really happens theoretically and illustrate with an experiment.
>
>
> **Convergence to true order under curve-orthogonal noise:**
> Thank you for this suggestion. We believe convergence is not possible even with purely curve-orthogonal noise when curvature is positive. Consider two nearby points on a curved section that looks like a bowl: moving orthogonally away from the curve causes their orthogonal lines to cross, after which projecting back to the curve reverses their order. As $n$ grows, this order reversal will affect a nonzero fraction of pairs (assuming $\Theta$ has continuous density), regardless of how small the orthogonal noise level is. This is a fundamental geometric limitation imposed by curvature, not a deficiency of any particular method.
>
>
>
> **About the choice of $\alpha$ in Section 5.1:** The choice of $\alpha$ was arbitrary after plotting some curves that seemed reasonable. We show additional results for $\alpha = 2$ and $\alpha = 3$ to demonstrate that the results are mostly the same.
>
> ### Comparison of Methods $(n = 2000)$ with $\alpha = 2$
>
> | Dim. | Method | $\sigma = 0.5$ | $\sigma = 1$ | $\sigma = 1.5$ | $\sigma = 2$ |
> |------|--------|----------------|--------------|----------------|--------------|
> | 50 | Fiedler Vector | 86.59 (2.19) | 77.77 (4.17) | 50.55 (15.26) | 22.86 (15.31) |
> | | 1D UMAP | 89.69 (0.67) | 78.86 (1.64) | 66.65 (2.91) | 50.03 (12.29) |
> | | 1D t-SNE | 91.27 (0.61) | 82.43 (1.32) | 73.29 (2.09) | 64.69 (3.44) |
> | | Recanati | 90.91 (0.82) | 81.5 (1.28) | 73.12 (1.64) | **66.21 (2.42)** |
> | | STAGE | **91.65 (0.6)** | **83.01 (1.35)** | **73.58 (2.18)** | 65.68 (2.95) |
> | 100 | Fiedler Vector | 75.9 (24.68) | 83.54 (1.87) | 68.47 (7.6) | 46.15 (10.92) |
> | | 1D UMAP | 91.88 (0.42) | 83.67 (0.96) | 73.89 (1.46) | 62.63 (2.28) |
> | | 1D t-SNE | 93.49 (0.31) | 87.03 (0.65) | 80.04 (0.99) | 72.03 (2.77) |
> | | Recanati | 93.34 (0.44) | 85.49 (0.33) | 79.6 (0.81) | **74.0 (0.85)** |
> | | STAGE | **93.9 (0.33)** | **87.6 (0.59)** | **80.78 (1.16)** | 73.79 (1.32) |
> | 200 | Fiedler Vector | 49.53 (21.36) | 89.05 (0.59) | 81.98 (0.89) | 69.2 (3.79) |
> | | 1D UMAP | 93.85 (0.22) | 87.63 (0.41) | 80.7 (0.75) | 73.08 (0.68) |
> | | 1D t-SNE | 95.43 (0.09) | 90.74 (0.29) | 86.11 (0.36) | 80.89 (0.56) |
> | | Recanati | 95.56 (0.16) | 88.7 (0.31) | 84.56 (0.29) | 80.45 (0.57) |
> | | STAGE | **95.85 (0.1)** | **91.47 (0.25)** | **86.86 (0.38)** | **82.24 (0.51)** |
>
> ### Comparison of Methods $(n = 2000)$ with $\alpha = 3$
>
> | Dim. | Method | $\sigma = 0.5$ | $\sigma = 1$ | $\sigma = 1.5$ | $\sigma = 2$ |
> |------|--------|----------------|--------------|----------------|--------------|
> | 50 | Fiedler Vector | 88.31 (1.04) | 74.8 (5.85) | 52.79 (11.41) | 22.32 (15.39) |
> | | 1D UMAP | 87.54 (0.97) | 75.45 (2.5) | 62.91 (3.17) | 48.5 (7.83) |
> | | 1D t-SNE | 89.59 (0.83) | 79.77 (1.85) | 70.3 (2.5) | 59.75 (3.45) |
> | | Recanati | 88.98 (1.3) | 78.83 (1.72) | 70.5 (1.96) | **62.36 (2.76)** |
> | | STAGE | **90.2 (0.8)** | **80.58 (1.73)** | **71.15 (2.6)** | 62.29 (3.13) |
> | 100 | Fiedler Vector | 89.25 (1.62) | 82.15 (2.37) | 60.52 (8.36) | 36.32 (8.56) |
> | | 1D UMAP | 90.09 (0.43) | 80.4 (1.68) | 68.8 (2.58) | 55.04 (5.92) |
> | | 1D t-SNE | 92.2 (0.46) | 84.33 (1.42) | 76.8 (1.5) | 67.8 (3.55) |
> | | Recanati | 91.9 (0.76) | 83.58 (1.13) | 76.43 (1.06) | 70.44 (1.78) |
> | | STAGE | **92.8 (0.47)** | **85.64 (1.0)** | **78.38 (1.54)** | **70.76 (2.27)** |
> | 200 | Fiedler Vector | 87.06 (4.21) | 88.34 (0.41) | 78.94 (1.8) | 63.35 (5.08) |
> | | 1D UMAP | 92.65 (0.29) | 85.07 (0.37) | 77.12 (0.95) | 67.45 (1.19) |
> | | 1D t-SNE | 94.51 (0.14) | 88.98 (0.27) | 83.44 (0.44) | 75.3 (4.82) |
> | | Recanati | 94.67 (0.19) | 87.42 (0.28) | 82.11 (0.49) | 77.24 (0.51) |
> | | STAGE | **95.02 (0.16)** | **90.0 (0.27)** | **84.94 (0.51)** | **79.74 (0.68)** |

---

### Official Review · Reviewer_EMVY · 2025-10-31

**Soundness:** 2
**Presentation:** 3
**Contribution:** 2
**Rating:** 2
**Confidence:** 3

**Summary:**

This paper introduces STAGE (Spectral Tangent Alignment and Geometric Embedding), a multi-stage algorithm designed to solve the linear seriation problem, which involves correctly ordering noisy data points sampled from a one-dimensional curve embedded in a high-dimensional space. The method first estimates the local tangent direction at each data point using Principal Component Analysis on its neighbors, then employs a spectral technique to ensure these tangent vectors are globally aligned in a consistent orientation. Following this alignment, the algorithm computes a final one-dimensional embedding by solving a least-squares problem that arranges the points according to this consistent directional flow, with the final order determined by sorting these embedded values. The authors demonstrate that STAGE is computationally efficient and often outperforms existing methods, particularly in high-dimensional and noisy settings, showing its effectiveness on some toy synthetic datasets and for pseudotime ordering in real-world single-cell RNA-seq data.

**Strengths:**

1. The paper's primary strength is its shift from traditional scalar-based methods to a novel vector-based approach. While algorithms like ISOMAP preserve distances and LLE preserves reconstruction weights, STAGE introduces the more dynamic concept of preserving the manifold's directional flow. Its core innovation lies in estimating local tangent vectors and, crucially, solving the non-trivial problem of aligning them into a globally consistent vector field, a challenge not addressed by the other methods.
2. The contribution is not simply a new heuristic in manifold learning but the authors support their algorithm with a solid theoretical analysis that provides guarantees on its ability to recover the correct ordering. This mathematical foundation, combined with its demonstrated computational efficiency and successful application to complex real-world problems like single-cell RNA-seq pseudotime ordering, makes the method both powerful and practical for researchers.

**Weaknesses:**

The paper has some weaknesses and I will try to write them down in a somewhat decreasing order of significance that would hopefully help the authors to fix these issues and improve the quality of their paper.

1. A notable weakness of the paper is its insufficient engagement with foundational methods in manifold learning, specifically ISOMAP and Locally Linear Embedding (LLE). While the algorithm is clearly inspired by this field, as acknowledged through a subtle citation to the original LLE paper, it stops short of providing a direct, quantitative comparison (e.g. page 2, lines 102-105). For a new method to establish its place, it is crucial to benchmark it against the very algorithms that represent the benchmarks like in distance preservation (ISOMAP) and in local reconstruction (LLE). By omitting such a direct comparison, the authors leave a critical gap in their analysis, making it difficult for readers to fully appreciate the practical performance gains and trade-offs of their vector-based approach relative to these well-established techniques.
2. Furthermore, the empirical validation of the STAGE algorithm is conducted on datasets of a relatively modest scale, with the largest example containing fewer than 17,000 genes. This scope raises significant questions about the method's scalability and its applicability to the large-scale "big data" problems prevalent today. An algorithm's performance on thousands of samples does not guarantee its feasibility on millions, where computational and memory constraints are really important. The paper would be substantially stronger if it explored how STAGE performs against modern, highly scalable embedding techniques, such as a simple autoencoder with some neural network architecture, which are designed to handle massive datasets efficiently and could potentially offer a more practical solution for a large number of samples.
3. The evaluation of the algorithm's success is also somewhat limited by its focus on the intrinsic quality of the ordering, rather than the extrinsic utility of the resulting embeddings. A truly powerful manifold learning algorithm should produce a low-dimensional representation that is useful for downstream machine learning tasks. A more robust validation strategy, as demonstrated in prior work like Paraskevopoulos et al. (2018) [A], would involve a task-based evaluation. For example, by feeding the embeddings generated by STAGE and other competing algorithms (such as SMACOF, LLE, and ISOMAP) into a standard k-NN classifier and comparing the resulting accuracies, one could obtain a clear, objective measure of how well each method preserves the essential structure of the data for a practical application.
4. Finally, I think the authors should include some computational complexity numbers of their algorithm vs the above referenced SOTA manifold learning algorithms to show maybe that their method is of similar or lower complexity (e.g. execution time or actual memory footprint)

[A] Paraskevopoulos, G., Tzinis, E., Vlatakis-Gkaragkounis, E.V. and Potamianos, A., 2018. Pattern search multidimensional scaling. arXiv preprint arXiv:1806.00416.

I would gladly increase my score if the authors work properly to address the above issues to a proper degree.

**Questions:**

Considering the novelty of STAGE for vector-based alignment, could you elaborate on its practical trade-offs against foundational methods like ISOMAP and LLE, specifically regarding how its computational complexity scales versus the utility of its embeddings for a downstream task (such as k-NN classification) when applied to large-scale datasets?

---

> ### Author Response · Authors · 2025-11-28
> **Response to comments**
>
> Thank you for your valuable feedback. Please find our answers to your comments below.
>
> **About comparison with LLE and ISOMAP:** Thank you for suggesting these methods. ISOMAP is similar to t-SNE in that both consider pairwise distances with neighbors and aim to preserve these distances in low-dimensional embeddings. We added experiments comparing with LLE and ISOMAP in Figures 9-11, in the revised Appendix E.2. The plots illustrate that our method is superior across various dimensions and sample sizes with $\sigma = 2$. Our runtime is slightly longer at lower sample sizes but better when $n = 2000$. We also found that LLE is very sensitive to the hyperparameter, while STAGE and ISOMAP are less sensitive. These results are in the "Higher Dimension Experiments" section of the appendix (Section E.2) of the revised paper.
>
> **About scale of the data:** The current three datasets are the most suitable we found for experiments. Our primary goal is applying this method to genomic data. We welcome suggestions for larger datasets.
>
> **About downstream task evaluation and k-NN classification:** We respectfully disagree with this suggestion. Our method is fundamentally designed for **seriation** (recovering a linear ordering of datapoints) not general manifold learning. The output is a 1-D ordering, not a multi-dimensional embedding for downstream tasks. Evaluating seriation quality via the intrinsic metric (Kendall's tau on the recovered order) is the direct and appropriate measure of success for this task, just as classification accuracy is the appropriate metric for classification methods.
>
> The suggested evaluation using k-NN classification on MNIST is not applicable: (1) our method produces only 1-D outputs, unsuitable for classification, and (2) our target datasets (particularly genomic data) lack categorical labels. Comparing STAGE to general manifold learning methods like LLE and ISOMAP on classification tasks would be comparing fundamentally different objectives. Our focus is linear seriation; generalization to manifold learning is future work.
>
> **About computational complexity:** The bottleneck is finding the first singular vector of $k$-nearest neighbors in $\mathbb{R}^d$ for all $n$ datapoints, with complexity $O(nkd)$. As mentioned above, our method is slower than ISOMAP and LLE (though it becomes faster than ISOMAP at larger sample sizes), but performs better in terms of order recovery.

---

### Official Review · Reviewer_gPKe · 2025-10-31

**Soundness:** 3
**Presentation:** 2
**Contribution:** 2
**Rating:** 4
**Confidence:** 3

**Summary:**

The paper proposes a three-step procedure for trajectory seriation consisting of: Local estimation, Alignment, 1D embedding, and provided a theoretical guarantee and simulations to show its superiorty over competing alogithms.

**Strengths:**

The paper presents a complete algorithmic framework, supported by simulations that demonstrate its advantages over competing methods.

**Weaknesses:**

While the structure of the approach is clear, the novelty of the contribution appears limited, as each step builds upon existing methods without a substantial new theoretical or algorithmic insight. The only novelty might be the step 3 of the "Constructing a Geometrically-Informed Global Embedding", but I am not sure how much improvement this step gives.

The statement of Theorem 1 is not clearly written and without discussion, hard to understand. The meanings of key quantities such as 𝑓_max and 𝐾_𝜅 are not explained, making it difficult to interpret the result. Moreover, the result does not seem to establish consistency, since the first term on the right-hand side is independent of n. The authors should clarify what the theorem is intended to demonstrate and, if possible, provide a discussion of the asymptotic behavior of the estimator.

To make the paper more compelling and easier to understand, it would be helpful to include a simple illustrative example. For instance, the authors could demonstrate their method on a one-dimensional manifold such as a straight line, a circle, or a half-circle. Such an example would concretely show how the proposed procedure works in practice and highlight its advantages or limitations.

**Questions:**

See weakness--
1. The meanings of key quantities such as 𝑓_max and 𝐾_𝜅 are not explained, making it difficult to interpret the result.
2. The result does not seem to establish consistency, since the first term on the right-hand side is independent of n. When does the theorem shows that the algorithm performs well with a small error bound on the RHS?
3. The authors should clarify what the theorem is intended to demonstrate and, if possible, provide a discussion of the asymptotic behavior of the estimator. It would be helpful to include a simple illustrative example. For instance, the authors could demonstrate their method on a one-dimensional manifold such as a straight line, a circle, or a half-circle.

---

> ### Author Response · Authors · 2025-11-28
> **Comment on novelty**
>
> Thank you for your valuable feedback. We will break up our responses to your comments into two parts due to space limitation. Let us first address the concern about novelty.
>
> **About the novelty of the paper:** We believe our method is novel in the following sense. We do not have innovations on local PCA. Each step is simple and straightforward (*except step 2, see the final comment on that*). However, we believe that the combination of the three main steps is novel. If you check most manifold learning or ordering papers, they have the following flavor:
>
> 1. Find $d_{ij} = \\|x_i-x_j\\|$.
>
> 2. Find $y_i$ in low dimensional space so that the distances are preserved: $\\|y_i-y_j\\|\approx d_{ij}$.
>
> These methods include t-SNE and ISOMAP as suggested by Reviewer EMVY.
>
> We want to emphasize that our approach is completely different, and to the best of our understanding, no paper has considered this approach before for similar problems.
>
> 1. Finding local coordinates for nearest neighbors.
>
> 2. Align the orientations for neighbors, *by globally aligning tangent spaces of a manifold*.
>
> 3. Find a global representation based on the local coordinate and the alignment.
>
> It is clear that our method has completely different flavor compared to existing methods. The main point is to avoid using $d_{ij}$ which loses information from original point cloud data. Our novelty is on the combination of these steps rather than innovation on a single step.
>
> One can also compare our approach with manifold learning methods. The goal of manifold learning targets recovery of the manifold, while the denoised location of a data point is not important. Hence, manifold learning does not develop or apply step 2 or 3 in our proposed method.
>
> Finally, we would like to note that casting the problem of aligning tangent spaces of a manifold as a ($\mathbb Z_2$) synchronization problem and solving it using ideas borrowed from that literature is "highly novel" to the best of our knowledge. It bridges two seemingly different literature/problems (manifold learning and synchronization). For example, it shows that synchronization is perhaps a more interesting and fundamental problem than the current literature imply.
>
> We have discussed these notions of novelty and differences in Section 2. We will add more discussion in a later revision.

---

> > ### Author Response · Authors · 2025-11-28
> > **Response to the rest of comments**
> >
> > **About simple illustrative example:** Some simple examples have already been presented in the original manuscript, such as Figure 1 and Figure 5. The labeled color of Figure 5 shows the order recovery of our algorithm. Original Figure 1 illustrates the effectiveness of orientation alignment. In the revised paper, we have added a third panel to Figure 1 to demonstrate the result of the full algorithm.
> >
> > > The meanings of key quantities such as $f_{\max}$ and $K_\kappa$ are not explained, making it difficult to interpret the result.
> >
> > Thank you for this feedback. We have added explanations of these quantities immediately after Theorem 1 and provided a comprehensive interpretation in Section 4.1. Please see the uploaded rebuttal revision.
> >
> > **Density bounds $f_{\min}$ and $f_{\max}$.** These are uniform lower and upper bounds on the density of $\Theta$ (defined in Section 4). They impose mild regularity conditions preventing the distribution from being too concentrated or too sparse. For example, the uniform distribution on $[0,1]$ has $f_{\max} = f_{\min} = 1$. The bound on $f_{\max}$ is essential: without it, MSE could be small while Kendall tau error remains large, since an arbitrarily concentrated distribution would allow large ranking changes with only small $\ell_2$ movements. (Note: We corrected a notational inconsistency in the revision, using $f_{\max}$ throughout instead of $f_{\Theta,\max}$.)
> >
> > **Curvature terms $\kappa$ and $K_\kappa$.** The maximum curvature $\kappa$ is defined in Assumption 1, and $K_\kappa$ (defined in the theorem statement) bounds the discrepancy between $\hat{W}\_{ij}$ and its ideal value $\theta_j - \theta_i$. Even with no noise and exact tangent vectors, curvature creates this gap, which $K_\kappa(2\rho + r)$ upper bounds. This term vanishes as $\kappa \to 0$. This clarification has been added after Theorem 1.
> >
> > > The result does not seem to establish consistency ...
> > > The authors should clarify what the theorem is intended to demonstrate ...
> >
> > Thank you for this important question. We have added Section 4.1 to clarify the interpretation.
> >
> > **Consistency is impossible with noise.** A key point is that consistency (driving $1-\tau$ to zero as $n \to \infty$) is fundamentally impossible in the presence of noise—not just for our method, but for any method. Even with infinite data, Kendall's tau cannot reach 1 when there is noise.
> >
> > To see why, consider a simple example: $\theta_i \in [0,1]$, $\epsilon_i \sim N(0, \sigma^2)$, and $x_i = \theta_i + \epsilon_i$ all in $\mathbb{R}$ (so no local PCA error). The noise swaps the order of a positive fraction of pairs, making them unorderable regardless of sample size. Increasing $n$ only helps with local PCA estimation—it cannot eliminate the fundamental difficulty from noise and curvature.
> >
> > **What the theorem shows.** In the noiseless, linear case ($\kappa = \sigma = 0$), Theorem 1 does establish consistency: the bound becomes $(d/n)^{1/3} + n^{-1/2} \to 0$ as $n \to \infty$ (with $d = o(n)$).
> >
> > With noise, the theorem shows that the error can be made arbitrarily small by controlling $\kappa$, $\sigma$, and $\rho$. Specifically, as $n \to \infty$:
> > $$
> > \lim_{n \to \infty} [1-\tau(\hat y,\theta - \bar \theta)] \lesssim \bigl(K_\kappa(2\rho + \sigma^2) + \rho^2 + \kappa^2(1 + \rho^4) + \rho\bigr)^{2/3}
> > $$
> > The right-hand side vanishes as the curvature and noise levels decrease, demonstrating that STAGE's error becomes arbitrarily small when the problem becomes easier (lower curvature and noise).

---

### Official Review · Reviewer_jA8B · 2025-11-05

**Soundness:** 3
**Presentation:** 2
**Contribution:** 2
**Rating:** 6
**Confidence:** 4

**Summary:**

This paper proposes STAGE, a geometric algorithm for linear seriation — the problem of recovering the intrinsic one-dimensional order of noisy samples drawn from an unknown curve embedded in high-dimensional space. The method proceeds in three main stages: (1) estimating local tangent directions via neighborhood PCA, (2) enforcing consistent global orientation of these tangents through a spectral relaxation of a sign-alignment objective, and (3) constructing a globally coherent 1-D embedding by solving a least-squares system based on the oriented tangents. Theoretical analysis establishes finite-sample recovery guarantees under bounded curvature and noise, providing probabilistic bounds on Kendall’s τ correlation with the ground-truth order. Empirical results on synthetic Fourier-curve data and single-cell RNA-seq pseudotime datasets show that STAGE achieves state-of-the-art rank recovery accuracy and runtime efficiency compared to spectral and manifold-learning baselines such as t-SNE, UMAP, and Recanati et al. (2018).

**Strengths:**

The paper is exceptionally well written and mathematically coherent, combining geometric intuition with rigorous analysis. Its key contribution is to operate directly on the point-cloud geometry rather than collapsing data into a global similarity matrix, thereby preserving fine-grained local structure. The use of local PCA to estimate tangents, followed by spectral sign alignment, is elegant and computationally efficient. The theoretical results are detailed and nontrivial, bridging tools from differential geometry, random matrix concentration, and spectral graph theory to derive rank-consistency bounds. The empirical section is solid and demonstrates that STAGE performs competitively across a wide range of noise levels and dimensions, often outperforming t-SNE and UMAP while being faster. The combination of geometric consistency, spectral reasoning, and provable robustness makes this work a valuable contribution to the intersection of manifold learning and seriation.

**Weaknesses:**

While theoretically sophisticated, the method’s scope is limited to one-dimensional manifolds. The authors acknowledge that the orientability assumption and the global sign-alignment procedure fail for higher-dimensional or non-orientable manifolds. Thus, the generalization of STAGE to k>1 manifolds remains open and would require synchronization over SO(k), which is nontrivial. Additionally, several technical assumptions — such as bounded curvature and strong connectivity of the neighborhood graph — may be difficult to verify in practical data. From an empirical standpoint, although the synthetic and biological experiments are convincing, it would be informative to test the algorithm on more diverse real-world domains (e.g., motion trajectories, time-series embeddings). Finally, while the theoretical analysis provides finite-sample guarantees, it focuses on asymptotic concentration without a detailed treatment of bias introduced by approximate tangent estimation, which might affect robustness at small sample sizes.

**Questions:**

1.	Extension to higher-dimensional manifolds.
Could the tangent-alignment stage be generalized to k>1 dimensional manifolds via frame synchronization over \mathrm{SO}(k)? For instance, do you see a viable path using connection Laplacians or orthogonal Procrustes–type relaxations, or are there conceptual obstacles beyond technical difficulty?
	2.	Sensitivity to noise and curvature / phase transition.
How sensitive is your theoretical guarantee to violations of the “small curvature / sub-Gaussian noise” assumptions? Do you expect (or empirically observe) a phase-transition–type behavior in Kendall’s \tau as curvature, noise level, or tube radius increase—i.e., a regime where recovery abruptly ceases to improve toward \tau \approx 1?
	3.	Asymptotic behavior of the main bound.
In the main theorem, could you comment on the asymptotic rate of each term in the upper bound as a function of n, curvature \kappa, noise level \sigma, tube radius \rho, and graph parameters (e.g., r, \lambda_2(L))? In particular, is it correct that, under your current assumptions, the first summand behaves like a (possibly large) constant determined by geometry and noise, so that only the n^{-1/2} term vanishes with growing sample size? Or is there a natural scaling of r, \rho, etc., under which the first term can also be driven to zero?

---

> ### Author Response · Authors · 2025-11-28
> **Response to comments**
>
> Thank you for your positive assessment and valuable feedback. Please find our answers to your questions below.
>
> **About extension to higher dimension manifolds:**
>
> It is in principle possible to extend our method to higher dimensional manifolds. The main challenge is developing an efficient and theoretically justified algorithm for the following variant of $O(k)$ synchronization:
>
> Given orthonormal frames $V_i \in \mathbb{R}^{d\times k}$ (columns orthonormal, approximately spanning the tangent space at point $i$), and find $O_i \in O(k)$ to maximize
>
> $$\max_{O_i \in O(k)} \sum_{i,j} A_{ij} \mathrm{tr} \left(O_i^\top V_i^\top V_j O_j\right).$$
>
> Although spectral relaxation might be possible by lifting to $nk \times nk$ matrices, theoretically analyzing such a method, under perturbation, is challenging, esp. as connected to the properties of the underlying manifold (more complex notation of curvatures, etc.). If this step can be overcome, there is a straightforward way to extend our method to achieve a higher dimensional embedding. One feature of that extension is that it would potentially preserve some notion of "order", perhaps along the curves on the manifold. Ironing out exactly what that notion is would be an interesting direction for future work.
>
>
> > How sensitive is your theoretical guarantee to violations of the “small curvature / sub-Gaussian noise” assumptions?
>
> We have added an expanded discussion of our theoretical bound in Section 4.1 of the revised manuscript.
>
> > Do you expect (or empirically observe) a phase-transition–type behavior in Kendall’s \tau a
>
>
> We have included a heatmap of our recovery versus noise $\sigma$ and curvature $\kappa$ in Figure 12 of the revised paper. This is for $\gamma$ being a portion of a circle embedded into $d= 10, 50, 500$-dimensional space, for $n = 1000, 2000$, included in the "Higher Dimension Experiments" section in Appendix E.2. Whether there is sharp phase transition, under a regime of scaling of $d$, $n$ and other features of the problem, is plausible.
>
> > Asymptotic behavior of the main bound
>
> Please see the added Section 4.1. Briefly, in the noisy case, one needs to take $r$ to be order 1 (it cannot go to zero); this implies that $\lambda_2(L)$ and the averge degree are both scaling like $n$, hence their ratio is constant and can be ignored. And yes most of the terms remain as $n \to \infty$ and this is expected since consistent recovery is not possible when
> $\sigma >0$ as discussed in that section. However, those terms do go to zero as $\kappa, \sigma \to 0$ (in which case $\rho$ can be taken to go to zero as well). So we can achieve arbitrarily small error, by requiring sufficiently small $\kappa$ and $\sigma$
>
> > ... method’s scope is limited to one-dimensional manifolds
>
> We aimed to formulate and solve the high-dimensional trajectory seriation problem in a principled way. This is inherently a one-dimensional problem. Our motivation was questions in bioinformatics and biostatistics which even lacked proper formulation. We believe, we are the first to propose to look and solve the problem in the original data domain, rather than taking the starting point a similarity matrix.
>
> It is true that our method provides an embedding and potentially can be generalized to define a new high-d embedding method. But more interesting, one could ask the question: What is the higher dimensional analog of seriation? Our method once extended would be solving such problem which yet does not exist!
>
> We believe the 1-dimensional case stands on its own a self contained contribution which could pave the way for these potentially more interesting extensions.
>
> It is worth noting that some of our empirical findings, such as the surprising performance of t-SNE for this task are novel on its own. We provide a new well-defined benchmark to measure the performance of existing and future embedding methods.
>
> > ... without a detailed treatment of bias introduced by approximate tangent estimation
>
> As part of proof of Theorem 1, we do provide such detailed analysis; some of the terms in the bound are coming exactly from this bias. Please see Lemma 4 and 5 in the appendix which together with Davis-Kahan lead to  an expression for the bias (the term $D_1$ divided by $f_{\min} (\delta)^3$ in (17) to be exact). This is also pointed out in the revision after Theorem 1 where we explain the sources of the various terms in the bound.

---

### Meta-Review · Area_Chair_YHmA · 2026-01-06

**Summary:**

This paper proposed a method for linear seriation to recover the intrinsic order of noisy samples from a curve embedded in high dimensional space. Four reviewers have diverse opinion on this with one positive and three negative. The authors resolved some concerns during the rebuttal but there are some remaining issues. Furthermore, there was an error in the main theorem (Theorem 1) in the initial submission, which requires further assumption. As a result, I can not recommend to accept. Here is a summary of unsolved major concerns:
1. Most reviewer found Section 4 not clearly written, with technical flaws. Although the authors revised Section 4, it reads beyond a minor revision but more like a major one.
2. The computational cost was not mentioned in initial submission, and later on claimed to be slower than Isomap and LLE. Note that Isomap is already considered slow in the literature. Further effort to speed it up is expected. Note that I don't fully agree with Reviewer EMVY's question about scalablity, who asked more examples with dimension more than tens of thousand and sample size at scale of million without a clear motivating example. However, I do think scalability needs to be considered and discussed more deeply in the next version.
3. For general audience, a natural question is what can we do with the recovered order? It's not discussed in deep in the paper, without a clear motivating real application. The paper will benefit from a better motivation at the beginning of the paper instead of directly starting with "We address the problem of linear seriation:..." I understand this is a methodology paper but even for purely theoretical paper a motivation is a big plus.

**Reviewer Concerns:**

Reviewer jA8B:
1. Extension to higher dimensional manifolds (not really addressed)
2. Sensitivity of theory to small curvature/sub-Gaussian noice assumptions (somewhat addressed by explanation in Section 4.1)
3. Missing asymptotics (partially addressed by Section 4.1)

Reviewer gPKe:
1. Novelty (partially addressed by highlighting the novelty in combing existing building blocks and avoid using pairwise distance, which may cause some loss of information)
2. Theorem 1 is not clearly written and explained (partially addressed by revising and expanding Section 4)
3. Need a simple illustrative example (addressed by expanding Figure 1)

Reviewer EMVY:
1. Missing baselines like Isomap, LLE, etc (addressed by adding experiments on LLE and Isomap)
2. Scalability concern (not directly addressed but I don't think the question is very reasonable)
3. Evaluation incomplete, esp for downstream tasks (not addressed by arguing the method is not manifold learning but seriation).
4. Missing computational complexity (complexity is provided, but the proposed method is slower than Isomap and LLE)

Reviewer LY8v:
1. An error in Theorem 1 and potential extension (seemingly addressed by adding more assumptions and fixing the proofs)
2. No sufficient support of the claim that similarity based methods are sensitive to outliers (partially addressed by some math derivation but not in formal theorem, and a simulation study)
3. Cherry picking of $\alpha$ in the simulation in Section 5.1 (addressed by further experiments for other $\alpha$s

**Reviewer Scores:**

Reviewer jA8B: 6 --> 6

Reviewer gPKe: 4 --> 5

Reviewer LY8v: 2 --> 2

Reviewer EMVY: 4 --> 4

---

### Decision · Program_Chairs · 2026-01-26

Reject